# DFRot: Achieving Outlier-Free and Massive Activation-Free for Rotated LLMs with Refined Rotation

**Jingyang Xiang**
New York University
xiangxiangjingyang@gmail.com

**Sai Qian Zhang**
New York University
sai.zhang@nyu.edu

## Abstract

Rotating the activation and weight matrices to reduce the influence of outliers in large language models (LLMs) has recently attracted significant attention, particularly in the context of model quantization. Prior studies have shown that in low-precision quantization scenarios, such as 4-bit weights and 4-bit activations (W4A4), randomized Hadamard transforms can achieve significantly higher accuracy than randomized orthogonal transforms. Notably, the reason behind this phenomenon remains unknown. In this paper, we find that these transformations show substantial improvement in eliminating outliers for common tokens and achieve similar quantization error. The primary reason for the accuracy difference lies in the fact that randomized Hadamard transforms can slightly reduce the quantization error for tokens with massive activations while randomized orthogonal transforms increase the quantization error. Due to the extreme rarity of these tokens and their critical impact on model accuracy, we consider this a long-tail optimization problem, and therefore construct a simple yet effective method: a weighted loss function. Additionally, we propose an optimization strategy for the rotation matrix that involves alternating optimization of quantization parameters while employing orthogonal Procrustes transforms to refine the rotation matrix. This makes the distribution of the rotated activation values more conducive to quantization, especially for tokens with massive activations. Our method enhances the Rotated LLMs by achieving dual free, *Outlier-Free* and *Massive Activation-Free*, dubbed as *DFRot*. Extensive experiments demonstrate the effectiveness and efficiency of DFRot. By tuning the rotation matrix using just a single sample, DFRot achieves a perplexity improvement of 0.98 and 0.95 on W4A4KV4 and W4A4KV16, respectively, for LLaMA3-70B, a model known for its quantization challenges. Code is available at https://github.com/JingyangXiang/DFRot.

## 1 Introduction

Large Language Models (LLMs) have shown exceptional abilities across numerous domains. Cutting-edge open-source models like LLaMA (Touvron et al., 2023) and Mistral (Jiang et al., 2023), along with proprietary LLMs such as GPT (Achiam et al., 2023) and Gemini (Team et al., 2023), are now being applied in a wide range of applications, including natural language understanding (Zellers et al., 2019; Hendrycks et al., 2020), machine translation (Zhang et al., 2023), content generation (Mo et al., 2024), recommendation systems (Wu et al., 2023; Wang et al., 2024; 2025) and agent (Li et al., 2025).

However, the remarkable success of LLMs is largely reliant on significant computational resources. LLMs often consist of billions of parameters, making them not only resource-intensive to train but also challenging to deploy on devices with limited computational capacity, such as mobile phones and edge devices. Additionally, the high memory and processing demands not only drive up hardware costs but also significantly increase energy consumption, leading to serious deployment concerns. To address these challenges, researchers and engineers are actively exploring various model compression techniques (Fran-

tar et al., 2022; Xiao et al., 2023; Lin et al., 2024a; Yao et al., 2022; Frantar & Alistarh, 2023; Ashkboos et al., 2024a; Wei et al., 2024; Zhao et al., 2025). These techniques aim to reduce the size of LLMs while maintaining their performance as effectively as possible, achieving a balance between efficiency and accuracy.

Unfortunately, the presence of outliers in the activations (Dettmers et al., 2022; Zeng et al., 2022) often leads to a significant reduction in model accuracy when PTQ is applied directly. To address this problem, earlier approaches have either scaled weights and activations (Xiao et al., 2023; Wei et al., 2023; Shao et al., 2023), shifting the quantization challenges from activations to weights, or employed mixed-precision techniques to isolate outliers (Dettmers et al., 2022), thereby minimizing the LLM's quantization error.

Recent research (Ashkboos et al., 2024b) has demonstrated that rotating activations in LLMs can effectively eliminate most outliers while preserving computational invariance, ensuring that the LLM's output remains identical to its original results. Moreover, the rotation matrices can be merged into the weights, imposing no additional burden on network inference. This innovative computational invariance (Ashkboos et al., 2024a) has garnered significant attention from researchers.

Although rotation is widely recognized as an important method for the quantization of LLMs, there remain many unresolved issues. For example, as shown in Table 1, when activations are reduced to 4-bit, the reasons why randomized Hadamard transforms (RH) often achieve significant improvement compared to randomized orthogonal transforms (RO) (Ashkboos et al., 2024b; Liu et al., 2024) have not yet been fully understood. However, while directly training rotation matrices can yield good results (Liu et al., 2024), the training process will cause substantial computational resources and adds complexity to the quantization process.

In this paper, we first investigate the underlying reasons why RH outperforms RO. We find that for ordinary tokens consisting primarily of outliers (Achiam et al., 2023), both RO and RH transformations can equally reduce quantization error when applied to these tokens. As shown in Figure 3, in terms of quantization error, there is no substantial difference between the two transformations. In contrast, for special tokens with *massive activations* (Sun et al., 2024), using RO on these activations surprisingly leads to an increase in quantization error. Our experiments show that this inability to efficiently manage massive activations greatly restricts the accuracy of quantized LLMs. On the other hand, while RH performs better than RO, it only manages to maintain or slightly reduce the quantization error for these large activations. This observation indicates that both transformation methods struggle to effectively manage massive activations in LLM quantization.

Building on these insights, we propose a novel optimization method to enhance the performance of quantized LLMs, achieving both *Outlier-Free* and *Massive Activation-Free, e.g.* dual free (*DFRot*). By treating scarce tokens with massive activations as long-tail distributed data, we develop a simple yet effective weighted loss function. Additionally, we introduce an alternating optimization approach to refine the rotation matrices and quantization parameters, further minimizing quantization error. Extensive experiments demonstrate the effectiveness of our proposed method. Specifically, by tuning the rotation matrix with just a single sample, DFRot achieves a PPL improvement of 0.95 and 0.98 on W4A4KV4 and W4A4KV16 for LLaMA3-70B with WikiText-2, a model recognized for its quantization challenges (Huang et al., 2024).

## 2 Related Work

### 2.1 Eliminating outliers via Scale Invariance

The initial idea behind suppressing outliers through scale invariance stems from the observation that weights are easier to quantize than activations, and outliers in activations often appear in a few fixed channels Dettmers et al., 2022. Based on this, SmoothQuant (Xiao et al., 2023) first proposes that we can offline migrate the quantization difficulty from activations to weights via scale invariance. SmoothQuant enables an INT8 quantization of both

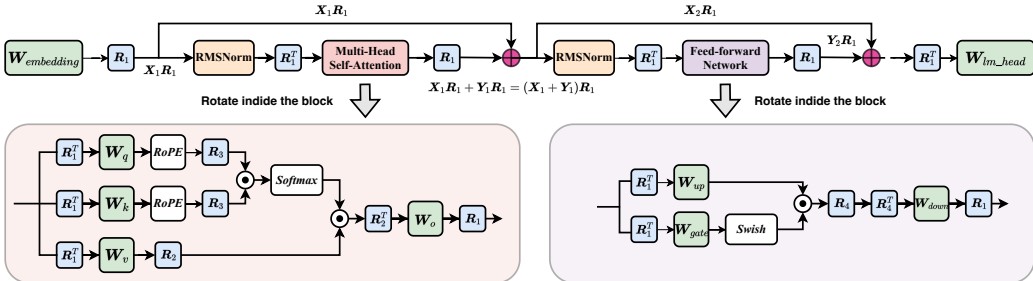

Figure 1: An illustration of rotational invariance in the LLaMA architecture. The rotation matrix $R_1$ can be integrated into the residual connection, ensuring the network retains rotational invariance. The rotation inner the block can further reducing outliers within block. Both of them make LLM fewer outliers and be easier to quantize. The rotation matrix $R_1$, $R_1^T$, $R_2$, $R_2^T$ and $R_4^T$ can be integrated into weights. $R_3$ and $R_4$ need to compute online.

weights and activations for all the matrix multiplications in LLMs. Furthermore, Outlier Suppression+ (Wei et al., 2023) proposes a fast and stable scheme to effectively calculate scaling values, achieving a better balance in quantization burden. To reduce manual design and further enhance quantization performance in extremely low-bit quantization, Omni-Quant (Shao et al., 2023) introduces Learnable Weight Clipping and Learnable Equivalent Transformation, efficiently optimizing the quantization process for both weight-only and weight-activation quantization. In the clipping W4A8 quantization, QQQ (Zhang et al., 2024) proposes to dynamically handle outliers through adaptive smoothing. QServe (Lin et al., 2024b) proposes SmoothAttention to effectively mitigate the accuracy degradation caused by 4-bit KV quantization. Both QQQ and QServe have effectively enhanced the performance of LLMs in W4A8 quantization.

## 2.2 Eliminating outliers via Rotational Invariance

Although scale invariance can reduce outliers and improve quantization performance, it merely transfers the outliers from activations to weights and has not eliminated them fundamentally. When the magnitude of the outliers is large, scaling struggles to achieve an effective balance between weights and activations. Recently, researchers have found that applying rotation matrices to networks can effectively reduce outliers without increasing the complexity of LLMs. QuIP Chee et al. (2024) is the first to suggest that quantization can benefit from the incoherence between weight and Hessian matrices. It employed randomized orthogonal matrices generated by Kronecker product to enhance their incoherence. QuIP# (Tseng et al., 2024) replaces the randomized orthogonal matrices with randomized Hadamard matrices, which are faster and possess better theoretical properties. QuaRot (Ashkboos et al., 2024b) is the first work to apply rotational invariance (Ashkboos et al., 2024a) for model quantization. QuaRot finds that randomized Hadamard transformations yield better results compared to randomized orthogonal transformations. SpinQuant (Liu et al., 2024) and OSTQuant (Hu et al., 2025) further extends the rotation matrices to a trainable space and applied Cayley optimization (Li et al., 2020) to refine them, achieving significant improvements across diverse datasets.

# 3 Rotational Invariance, Quantization and Massive Activation

## 3.1 Rotational Invariance

First, we briefly introduce rotational invariance in LLMs, using the structure of LLaMA as an example. We assume that the $\alpha$ in the RMSNorm has been fused into the follow linear layers' weights, including $W_q$, $W_k$, $W_v$, $W_{up}$ and $W_{gate}$ and RMSNorm applies to each row of the activations $X$ as $X_{i,:} \leftarrow X_{i,:} / \|X_{i,:}\|$. If $R_1$ is an rotation matrix, we have the commutation property $\text{RMSNorm}(X R_1) = \text{RMSNorm}(X) R_1$ (Ashkboos et al., 2024a). This property implies that multiplying the input of RMSNorm by $R_1$ is equivalent to multiplying the RMSNorm output by $R_1$.

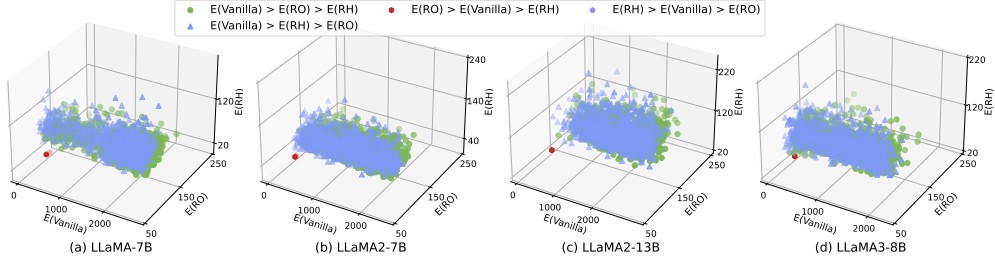

Figure 2: Comparison of 4-bit activation quantization error $E(\cdot)$ for each token with NR, RO and RH for (a) LLaMA-7B, (b) LLaMA2-7B, (c) LLaMA2-13B and (d) LLaMA3-8B. The tokens are from model.layers.6.post_attention_layernorm. Best viewed in color.

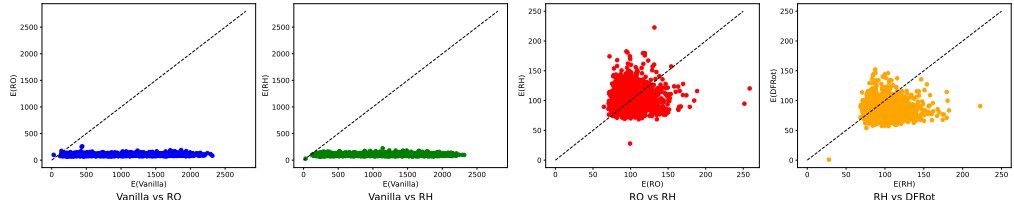

Figure 3: Comparison of 2D 4-bit quantization errors for tokens with NR, RO, RH and DFRot for LLaMA3-8B from Figure 2.

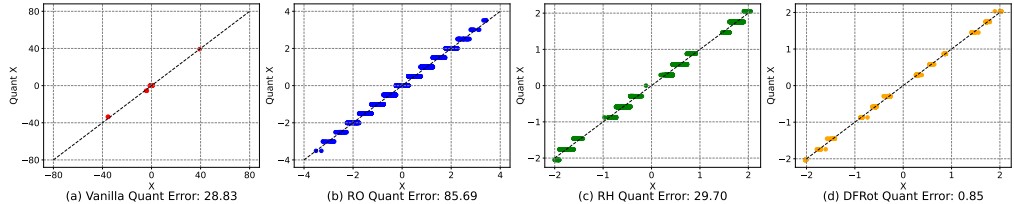

Figure 4: Comparison of 4-bit quantization error for the token with massive activation with NR, RO, RH and DFRot for LLaMA3-8B from Figure 2.

As shown in Figure 1, to remove outliers in the input activations, a rotation matrix $R_1$ is applied to the embedding layer $W_{embedding}$, resulting in a new input activation $X_1 R_1$. According to the above, we can know once we transform $W_q$, $W_k$, $W_v$ and $W_o$ in the Multi-Head Attention (MHA) to $R_1^T W_q$, $R_1^T W_k$, $R_1^T W_v$ and $W_o R_1$, the hidden feature within the MHA will remain unchanged, and the original output feature $Y_1$ will become $Y_1 R_1$. The following Feed-Forward Network's input $X_2$ from the residual connection will be modified to $(X_1 + Y_1)R_1 = X_2 R_1$. If we further transform $W_{up}$, $W_{gate}$ and $W_{down}$ to $R_1^T W_{up}$, $R_1^T W_{gate}$ and $W_{down} R_1$, the hidden feature within the FFN will also remain unchanged, and the output feature $X_3$ will be modified to $(X_2 + Y_2)R_1 = X_3 R_1$. Based on mathematical induction, we can get that $X_n R_1 + Y_n R_1 = (X_n + Y_n)R_1 = X_{n+1} R_1$ for the $n$-th module. To this end, by transforming $W_{lm\_head}$ into $R_1^T W_{lm\_head}$, the network output will remain unchanged.

There is also rotational invariance within the block. For MHA, we can insert head-wise rotation matrices $R_2$ and $R_2^T$ for $W_v$ and $W_o$ and $R_3$ for **Query** and **Key** after RoPE. For FFN, we can insert $R_4$ and $R_4^T$ between Swish and $W_{down}$. These approaches can further eliminate outliers and reduce quantization error while keeping the block output unchanged. In this paper, we only discuss $R_1$. For $R_2$, $R_3$, and $R_4$, we follow the QuaRot (Ashkboos et al., 2024b) settings and use Hadamard matrices.

## 3.2 Why the Randomized Hadamard is better than Randomized Orthogonal?

Based on the computational invariance described in Section 3.1, it is evident that the choice of rotation matrices is critical for ensuring the accuracy performance of the quantized model. Therefore, a natural question arises: **What type of rotation matrix offers the most advantageous properties?**

| Method | LLaMA-7B | | | LLaMA2-7B | | | LLaMA2-13B | | | LLaMA3-8B | | |
|---|---|---|---|---|---|---|---|---|---|---|---|---|
| | 4-4-4 | 4-4-16 | 4-8-16 | 4-4-4 | 4-4-16 | 4-8-16 | 4-4-4 | 4-4-16 | 4-8-16 | 4-4-4 | 4-4-16 | 4-8-16 |
| GPTQ | NaN | NaN | NaN | NaN | NaN | NaN | Inf | Inf | 6.01 | Inf | Inf | 7.29 |
| (RO) QuaRot | 6.68 | 6.62 | 5.80 | 7.96 | 7.71 | 5.61 | 6.00 | 5.92 | 4.99 | 10.54 | 10.15 | 6.52 |
| (RO) QuaRot.FP16() | **6.30** | **6.27** | - | **6.17** | **6.10** | - | **5.38** | **5.34** | - | **7.83** | **7.68** | - |
| (RH) QuaRot | 6.37 | 6.33 | 5.81 | 6.27 | 6.20 | 5.61 | 5.51 | 5.46 | 5.01 | 8.20 | 8.02 | 6.52 |
| (RH) QuaRot.FP16() | **6.30** | **6.28** | - | **6.17** | **6.10** | - | **5.40** | **5.37** | - | **7.82** | **7.67** | - |

Table 1: WikiText-2 perplexity (↓) results for RO and RH for LLaMA models. The 4-4-4, 4-4-16, 4-8-16 represent W4A4KV4, W4A4KV16, W4A8KV16 respectively. We show the failed GPTQ using NaN and the perplexity results>100 by Inf. QuaRot.FP16() denotes retaining tokens with massive activations as FP16.

We begin by focusing on RO and RH, as both QuaRot (Ashkboos et al., 2024b) and Spin-Quant (Liu et al., 2024) have demonstrate that RH delivers substantial improvements over RO in LLMs. We conducted experiments by applying RO and RH to the LLaMA models respectively, followed by weight quantization using GPTQ under various quantization settings. The results are shown in Table 1. Benefiting from the outlier elimination through rotational invariance, we find that for dynamical token-wise 8-bit activation quantization, both RO and RH lead to significant performance improvements compared to standard quantization. Additionally, no substantial performance difference is observed between the two transformations. **However, under 4-bit dynamical token-wise activation quantization, RH significantly outperforms** RO.

To investigate the performance differences between RH and RO under 4-bit activation setting, we plot the corresponding quantization error after applying 4-bit quantization to the multiple tokens. We also display the quantization error for the baseline setting where quantization is applied without rotating the activation to better understand the impact of using the rotation matrix. As shown in Figure 2, compared to the no rotation (NR), both RO and RH effectively reduce the quantization error for most tokens across different models. While RH slightly lowers the quantization error, the difference between the two methods is minimal for the majority of tokens. This leads to the question: **What explains the significant difference in PPL during quantization when their quantization errors are so similar?**

To answer this question, we turn our attention to massive activation (Sun et al., 2024), a rare but significant feature in LLMs. As shown in Figure 2, the red points represent quantization error for the tokens with massive activation. While most tokens show large quantization errors under NR, these special tokens display significantly smaller errors, which can be observed from Figure 3. It is normal since each token has a fixed $L_2$ norm after RMSNorm processing, as shown in Figure 4(a), tokens with massive activation naturally exhibit smaller quantization errors when quantized to 4-bit. Figure 4 presents the quantization result for the token with massive activation after applying NR, RO, and RH. Surprisingly, the rotation operations do not significantly reduce quantization errors for these tokens. In fact, compared to NR, RO greatly increases their quantization error, while RH only marginally reduces it. **This leads us to question whether tokens with massive activation are the primary cause of the significant accuracy discrepancies between** RH **and** RO.

To investigate this further, we build upon QuaRot by retaining tokens with massive activations in FP16 format for both RO and RH, while applying 4-bit quantization to the remaining input tokens, denoted as (RO) QuaRot.FP16() and (RH) QuaRot.FP16(). As shown in Table 1, for all LLaMA models, the performance gap between RH QuaRot and RO QuaRot is totally disappeared. It is so surprising that by simply retaining these extremely few tokens (often less than one-thousandth) as FP16, we can completely eliminate the performance difference between RO and RH. Therefore, we can make the following conclusion:

> **Why the Randomized Hadamard is better than Randomized Orthogonal?**
>
> RH = RO + Tokens with Massive Activations: RH is better than RO because it performs more effectively when reducing the quantization error for tokens with massive activations in 4-bit activation quantization.

### 3.3 Optimization Objectives and Calibration Data Selection

As mentioned above, although retaining tokens with massive activations as high-precision floating-point numbers can significantly enhance model accuracy, this approach is akin to a token-level version of LLM.int8(). It still requires fine-grained mixed-precision computations during the process, which will introduce additional system level optimization. Therefore, in this paper, we focus on W4A4 quantization to maintain simplicity and efficiency in the computation process. We consider a loss function of the following form:

$$\mathcal{L}(R_1, g_x) = \mathbb{E}_x \left[ \|xR_1 - \mathcal{Q}(xR_1, g_x)\|_2^2 \right], \tag{1}$$

where $x \in \mathbb{R}^{1 \times C}$ is the token vector from a calibration dataset $X^{cal} \in \mathbb{R}^{L \times C}$. $C$ is the hidden size and $L$ is the number of tokens. $R_1 \in \mathbb{R}^{C \times C}$ satisfies $R_1 R_1^T = I$, $g_x$ is the quantization parameters and $\mathcal{Q}(x, g_x) \in \mathbb{R}^{1 \times C}$ is the quantization of the $x$. The expectation $\mathbb{E}[\cdot]$ is taken over the token distribution. For the ease of analysis, we use the mean squared error $\| \cdot \|_2$.

Meanwhile, we introduce our data selection principle. We denote the calibration dataset as $X$, the tokens with massive activations as $X^m$, and the remaining tokens as $X \setminus X^m$:

$$\mathcal{L}(R_1, g_x) = \mathbb{E}_{x \in X^{cal} \setminus X^m} \left[ \|xR_1 - \mathcal{Q}(xR_1, g_x)\|_2^2 \right] + \gamma^2 \mathbb{E}_{x \in X^m} \left[ \|xR_1 - \mathcal{Q}(xR_1, g_x)\|_2^2 \right]. \tag{2}$$

During calibration, we apply a weighted loss to prioritize the quantization error on tokens with massive activations, with $\gamma$ representing the weight.

The motivation behind this principle stems from the observations in Table 1. Since $X^m$ is the key factor contributing to the performance gap between RO and RH, simply optimizing $R_1$ via Eq. 1 fails to specifically target $X^m$. On the other hand, compared to the NR in Table 1, RO also significantly improves performance, indicating that reducing the outliers on $X^{cal} \setminus X^m$ can enhance quantization performance, optimizing only for $X^m$ has the risk of increasing the quantization error for $X^{cal} \setminus X^m$, ultimately degrading the model's performance. Hence, it is crucial to optimize both $X^m$ and $X^{cal} \setminus X^m$. Naturally, we can regard this a long-tail optimization problem, where $X^m$ represents the long-tail but important data. Using a weighted approach to optimize the quantization loss is a simple yet highly effective method. Ablation studies in Section 4.2 further demonstrate the advantages of this strategy.

### 3.4 Solution Methods

Optimizing $R_1$ is a challenging task. Since $R_1$ influences every MHA and FFN in the network, adjusting the activation distribution in one layer impacts the quantization results across all layers. This makes it difficult to optimize layer by layer or block by block (Shao et al., 2023; Wei et al., 2023). A straightforward approach is to use training methods for quantization-aware fine-tuning of the rotation matrix across the entire network (Liu et al., 2024). Although it does not require retaining the gradients of the weights or the corresponding optimizer states, it still demands substantial computational resources during the quantization process.

In this paper, we focus on improving the effectiveness of rotation matrices in mitigating outliers and massive activation. Intuitively, we hypothesize that a rotation matrix that minimizes quantization error will lead to better performance. Drawing inspiration from Simsiam (Chen & He, 2021), we propose to regard quantization representation $\mathcal{Q}(xR_1, g)$ as cluster centroids $\eta_x$. In the context, optimizing $R_1$ and $g$ is equivalent to optimizing $R_1$ and $\eta_x$, which can be viewed as an implementation of an Expectation-Maximization (EM)-like algorithm, as shown in the following equation:

$$\min_{R_1, \eta_x} \mathcal{L}(R_1, \eta_x) = \mathbb{E}_{x \in X^{cal} \setminus X^m} \left[ \|xR_1 - \eta_x\|_2^2 \right] + \gamma^2 \mathbb{E}_{x \in X^{cal}} \left[ \|xR_1 - \eta_x\|_2^2 \right]$$
$$= \mathbb{E}_{x \in \widehat{X}^{cal}} \left[ \|xR_1 - \eta_x\|_2^2 \right], \tag{3}$$

where $\eta_x = \mathcal{Q}(xR_1, g)$ and $\widehat{X}^{cal} = \{x | x \in x \in X^{cal} \setminus X^m\} \cup \{\gamma x | x \in X^m\}$. This formulation is analogous to k-means clustering (Macqueen, 1967), and $R_1$ and $\eta_x$ act like the kernel

function and cluster centroids, respectively. Similar to k-means clustering, the problem described in Eq 3 can be approached using an alternating algorithm, where one set of variables is fixed while solving for the other. Formally, we can alternate between solving these two subproblems:

$$\eta_x^t \leftarrow \arg\min_{\eta_x} \mathcal{L}\left(R_1^{t-1}, \eta_x\right); \quad R_1^t \leftarrow \arg\min_{R_1} \mathcal{L}\left(R_1, \eta_x^t\right) \tag{4}$$

where $t$ represents the iteration index of the alternating rounds, and $\eta_x^t$ and $R_1^t$ denote the values of $\eta_x$ and $R_1$ at round $t$.

**Solving for the cluster centroids $\eta_x$.** The set of quantization parameters $g_x$ further contains the quantization scale $s_x$ and zero point $z_x$. In this paper, we adopt dynamic asymmetric per-token quantization for activations. Therefore, we can independently determine the optimal quantization scheme for solving $s_x$ and $z_x$ for each $xR_1$:

$$\eta_x = \mathcal{Q}_g(xR_1, s_x^t, z_x^t) = \text{clamp}\left(\left\lfloor \frac{xR_1}{s_x} \right\rceil + z_x, 0, 2^N - 1\right),$$

$$\text{where } s_x = \frac{\max(xR_1) - \min(xR_1)}{2^N - 1}, z_x = -\left\lfloor \frac{\min(xR_1)}{s_x} \right\rceil \tag{5}$$

where $\lfloor \cdot \rceil$ indicates round operation, $N$ is the bitwidth.

**Solving for $R_1$.** The right side od Eq 4 is well-known as Procrustes problem (Mulaik, 2009). which involves finding the optimal rotation matrix $R_1$ that best aligns two sets of points, minimizing the Frobenius norm of their difference. The solution to this problem can be obtained through Singular Value Decomposition (SVD). Specifically, given input matrices $X$ and its quantized version $\mathcal{Q}(X, g)$, the optimal $R_1$ can be found:

$$R_1 = UV^T, \text{where } U, \Sigma, V^T = \text{SVD}(X^T \mathcal{Q}(X, g_x)). \tag{6}$$

where we treat the quantization parameters $g^t$ as a constant.

**One-step optimization.** To find an improved rotation matrix $R_1$ and quantization parameters $g_x$, we perform the iterative process shown in Eq 4. Specifically, a calibration set $X^{cal}$ is randomly sampled from $X$, the iterative process can be specified as:

$$s_x^t, z_x^t \leftarrow \arg\min_{s_x, z_x} \sum_{x \in X^{cal}} \left[\left\| xR_1^{t-1} - \mathcal{Q}_{s,z}(xR_1^{t-1})\right)\right\|_2^2\right], \eta_x^t \leftarrow \mathcal{Q}_{s^t, z^t}(xR_1^{t-1}), \tag{7}$$

then the resulting quantization parameters will be used to produce the rotation matrix:

$$R_1^t \leftarrow \arg\min_{R_1} \sum_{x \in X^{cal}} \left[\left\| xR_1 - \eta_x^t\right\|_2^2\right] \tag{8}$$

The detailed algorithm is provided in Algorithm 1.

## 4 Experiments

**Experiment settings.** We implemented DFRot based on QuaRot. In this paper, to simplify the problem, we apply dynamic asymmetric per-token quantization for activation values. The KV-cache is quantized using asymmetric quantization with a group size of 128. GPTQ (Frantar et al., 2022) are used for weight with per-channel symmetric quantization, where a linear search for the clipping ratio is applied to minimize squared error. We use a sample with sequence length of 2048 from WikiText-2 (Merity et al., 2016) training set to genrate calibration dataset $X^{cal}$, initialize the rotation matrix $R_1$ with RH, and optimize it for 100 iterations. After obtaining the optimized rotation matrix $R_1$, we apply it to the corresponding model and achieve rotational invariance. We use 128 samples each with a sequence length of 2048, as the calibration dataset for GPTQ quantization.

| #Bits W-A-KV | Method | LLaMA3-8B | | LLaMA3-70B | | LLaMA2-7B | | LLaMA2-13B | | LLaMA2-70B | | LLaMA-7B | | LLaMA-13B | | LLaMA-30B | |
|---|---|---|---|---|---|---|---|---|---|---|---|---|---|---|---|---|---|
| | | 0-shot$^9$ Avg.($\uparrow$) | Wiki ($\downarrow$) | 0-shot$^9$ Avg.($\uparrow$) | Wiki ($\downarrow$) | 0-shot$^9$ Avg.($\uparrow$) | Wiki ($\downarrow$) | 0-shot$^9$ Avg.($\uparrow$) | Wiki ($\downarrow$) | 0-shot$^9$ Avg.($\uparrow$) | Wiki ($\downarrow$) | 0-shot$^9$ Avg.($\uparrow$) | Wiki ($\downarrow$) | 0-shot$^9$ Avg.($\uparrow$) | Wiki ($\downarrow$) | 0-shot$^9$ Avg.($\uparrow$) | Wiki ($\downarrow$) |
| 16-16-16 | FloatingPoint | 68.09 | 6.14 | 73.81 | 2.86 | 65.21 | 5.47 | 67.61 | 4.88 | 71.59 | 3.32 | 64.48 | 5.68 | 66.67 | 5.09 | 70.00 | 4.10 |
| 4-4-16 | RTN | 33.42 | 6e2 | 31.21 | 8e3 | 32.44 | nan | 30.86 | 8e3 | 30.90 | 7e4 | 32.51 | 7e3 | 31.63 | 3e4 | 31.57 | 2e3 |
| | SmoothQuant | 33.04 | 1e3 | 34.67 | 2e2 | 32.13 | nan | 34.26 | 1e3 | 35.86 | 3e2 | 34.42 | 3e2 | 33.29 | 6e2 | 34.64 | 1e3 |
| | GPTQ | 32.98 | 5e2 | 31.47 | 4e4 | 32.72 | nan | 30.11 | 4e3 | 30.86 | nan | 32.12 | 1e3 | 31.51 | 3e3 | 30.88 | 2e3 |
| | QuaRot | 61.86 | 8.11 | 68.25 | 5.92 | 61.63 | 6.17 | 64.66 | 5.45 | 69.96 | 3.89 | 61.65 | 6.33 | 64.83 | 5.57 | 67.79 | 4.74 |
| | DFRot | 63.01 | 7.78 | 69.82 | 4.97 | 62.42 | 6.13 | 65.34 | 5.39 | 69.16 | 3.99 | 62.25 | 6.30 | 64.47 | 5.58 | 68.06 | 4.78 |
| | SpinQuant* | 64.11 | 7.28 | 66.99 | 6.10 | 57.37 | 6.78 | 63.23 | 5.24 | 70.58 | 3.68 | 61.82 | 6.08 | 64.59 | 5.36 | 68.08 | 4.53 |
| | OSTQuant* | 65.14 | 7.24 | 72.21 | 3.97 | 63.90 | 5.60 | 66.24 | 5.14 | 70.92 | 3.57 | 62.72 | 6.04 | 65.80 | 5.40 | 68.52 | 4.43 |
| 4-4-4 | RTN | 33.18 | 7e2 | 30.82 | 8e3 | 32.67 | nan | 30.93 | 7e3 | 31.73 | 7e4 | 32.87 | 1e4 | 31.33 | 3e4 | 31.64 | 2e3 |
| | SmoothQuant | 32.96 | 1e3 | 33.76 | 3e2 | 32.12 | nan | 33.36 | 1e3 | 35.54 | 3e2 | 33.32 | 3e2 | 33.28 | 5e2 | 34.65 | 1e3 |
| | GPTQ | 33.71 | 6e2 | 31.20 | 4e4 | 33.52 | nan | 27.85 | 5e3 | 31.09 | nan | 31.80 | 2e3 | 30.63 | 3e3 | 31.07 | 2e3 |
| | Omniquant | 32.33 | 4e2 | - | - | 48.40 | 14.26 | 50.35 | 12.30 | - | - | 48.46 | 11.26 | 45.63 | 10.87 | 45.04 | 12.35 |
| | QuaRot | 61.38 | 8.28 | 68.29 | 6.02 | 60.81 | 6.25 | 64.44 | 5.49 | 69.96 | 3.92 | 61.21 | 6.37 | 64.68 | 5.59 | 67.92 | 4.77 |
| | DFRot | 62.94 | 7.91 | 69.62 | 5.03 | 61.80 | 6.25 | 64.95 | 5.43 | 68.78 | 4.02 | 61.84 | 6.36 | 64.26 | 5.62 | 67.93 | 4.81 |
| | SpinQuant* | 64.10 | 7.35 | 66.31 | 6.24 | 62.01 | 5.96 | 64.13 | 5.74 | 70.57 | 3.61 | 61.32 | 6.12 | 64.95 | 5.39 | 68.14 | 4.55 |
| | OSTQuant* | 65.37 | 7.29 | 71.69 | 4.01 | 63.18 | 5.91 | 65.41 | 5.25 | 70.84 | 3.59 | 62.55 | 6.07 | 65.43 | 5.40 | 68.20 | 4.42 |

Table 2: Comparison of averaged accuracy on nine Zero-Shot tasks and perplexity on WikiText2. Results for SmoothQuant, GPTQ, OmniQuant, AWQ, SpinQuant and OSTQuant are from the OSTQuant paper, and QuaRot's results from the official code. * denotes the methods that use the quantization-aware training to optimize $R_1$.

## 4.1 Main results

**Language Generation Task.** We evaluate DFRot on a language generation task and compare it with SmoothQuant (Xiao et al., 2023), GPTQ (Frantar et al., 2022), OmniQuant (Shao et al., 2023), AWQ (Lin et al., 2024a), SpinQuant (Liu et al., 2024) and OSTQuant (Hu et al., 2025). Table 2 shows the perplexity of LLaMA models. As shown, compared to QuaRot, DFRot achieves improvements in most cases. For example, DFRot achieves the most significant improvement on the LLaMA3-8B model with W4A4KV4 and W4A4KV16, outperforming QuaRot by 0.25 and 0.21, respectively. It is worth noting that DFRot has achieved near 1.00 PPL improvement on LLaMA3-70B, a model known for its challenging quantization performance, even surpassing SpinQuant, which finetunes $R_1$ on wikitext through quantization-aware-training.

Similar to QuaRot, DFRot does not require any retraining process and only needs a sample to optimize the rotation matrix. On a single NVIDIA A100 80G GPU, it only takes an extra 8 minutes for LLaMA-7B & LLaMA2-7B & LLaMA3-8B and 20 minutes for LLaMA2-13B, resulting in minimal overhead. Even for the 70B models, the additional time is less than 90 minutes, which is also acceptable. It demonstrates that DFRot has wide applicability and can serve as a cost-effective post-training method to enhance the quantization performance of rotated LLMs. Although DFRot does not achieve the best performance compared to the state-of-the-art methods, like OSTQuant, we believe DFRot also help community to understand the fundamental performance gap between RO and RH.

**Zero-Shot Tasks.** We also evaluate DFRot on the following nine important zero-shot tasks: BoolQ (Clark et al., 2019), PIQA (Bisk et al., 2020), WinoGrande (Sakaguchi et al., 2021), OpenBookQA (Mihaylov et al., 2018), SIQA (Sap et al., 2019), HellaSwag (Zellers et al., 2019), Arc (Easy and Challenge) (Clark et al., 2018) and LAMBADA (Radford et al., 2019). We use lm_eval==0.4.5 (Gao et al., 2024) or our experiments. Table 2 shows the average score of DFRot on the above tasks. As can be seen, DFRot consistently achieves improvements compared to QuaRot across all tasks. For example, DFRot achieves a 1.56% accuracy improvement compared to QuaRot on the LLaMA3-8B model with W4A4KV4 quantization settings.

## 4.2 Ablation studies

**Choice of $\gamma$.** To further understand the effect of hyperparameters in DFRot, we conducted an ablation study on Wikitext-2 PPL to investigate the impact of different $\gamma$ settings for W4A4KV16. As seen in Figure 5, when $\gamma$ ranges between 50 and 200, DFRot achieves significant improvements across various LLaMA models using RH. Notably, on the LLaMA3-8B

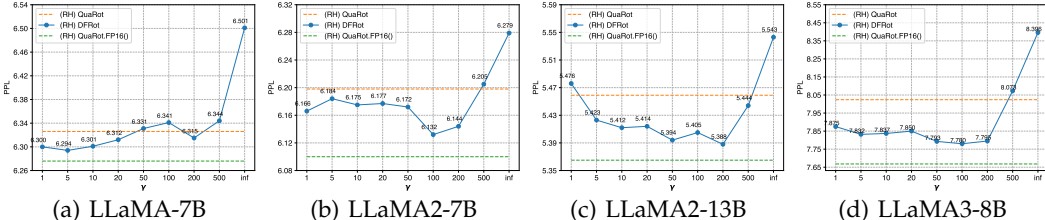

Figure 5: Comparison of WikiText-2 perplexity results under different $\gamma$ for W4A4KV16. $\boldsymbol{R}_1$ is initialized with RH.

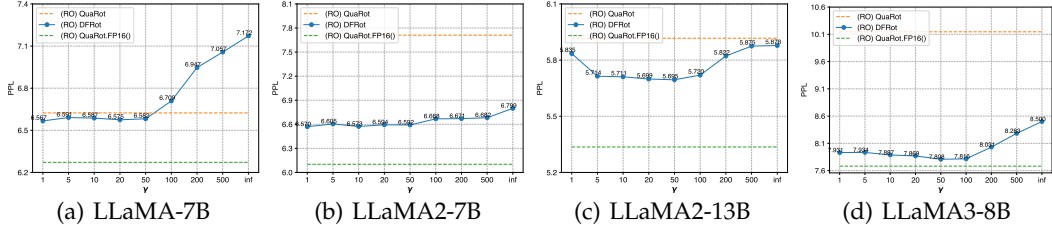

Figure 6: Comparison of WikiText-2 perplexity results under different $\gamma$ for W4A4KV16. $\boldsymbol{R}_1$ is initialized with RO.

model, which is known for its quantization performance sensitiveness to massive activations from Table 1, we observed a PPL improvement of over 0.2 in Figure 5(d). If we set $\gamma = 1$ and treat $\boldsymbol{X}^m$ and $\boldsymbol{X}^{cal} \setminus \boldsymbol{X}^m$ equally to minimize their quantization errors, it may reduce the quantization loss of $\boldsymbol{X}^{cal} \setminus \boldsymbol{X}^m$ but increase the quantization loss of $\boldsymbol{X}^m$, ultimately resulting in a performance decline on the LLaMA2-13B. Conversely, if we set $\gamma \to \infty$ and only optimize the quantization error for $\boldsymbol{X}^m$, it will increase the quantization error of $\boldsymbol{X}^{cal} \setminus \boldsymbol{X}^m$, resulting in an accuracy drop across the LLaMA-7B, LLaMA2-7B, LLaMA2-13B and LLaMA3-8B models.

**Initialize with Randomized Orthogonal.**   We conducted an ablation to study the effectiveness of DFRot when $\boldsymbol{R}_1$ initialized with RO. We keep the same experimental settings as in the study with RH and optimize the rotation matrix with different $\gamma$ values. As shown in Figure 6, our method achieves considerable improvements in RO scenarios compared to using RH for initialization. Meanwhile, it is more effective for LLM whose quantization peformance is more sensitive to the massive activations, such as LLaMA3-8B and LLaMA3-70B. However, due to the exceptional performance of RH, initialization and optimization using RH always yield superior final results compared to those obtained with RO. Therefore, we recommend using RH for initialization in practice to achieve better performance.

| Model | Sample1 (64×2048) | Sample2 (64×2048) | Sample3 (64×2048) | Sample4 (64×2048) | Sample5 (64×2048) |
|---|---|---|---|---|---|
| LLaMA3-8B | 7.78 | 7.76 | 7.79 | 7.74 | 7.76 |
| LLaMA2-7B | 6.13 | 6.12 | 6.15 | 6.11 | 6.14 |
| Model | Sample1 (48×2048) | Sample1 (32×2048) | Sample1 (24×2048) | Sample1 (16×2048) | Sample1 (8×2048) |
| LLaMA3-8B | 7.78 | 7.78 | 7.77 | 7.80 | 7.86 |
| LLaMA2-7B | 6.14 | 6.15 | 6.15 | 6.12 | 6.20 |

Table 3: Comparison of WikiText-2 perplexity results under different calibration samples for W4A4KV16.

## 4.3   Analysis of Calibration Set Sensitivity

We performed ablation studies W4A4KV16 on LLaMA3-8B and LLaMA2-7B along two dimensions: the choice of calibration samples and the num of calibration tokens and evaluate

results on WikiText. Samples are all sampled from WikiText-2 train. In selecting the number of tokens, we utilize the inputs to the first N transformer blocks as the calibration data source and demonstrate results in Table 3. For example, when $48 \times 2048$ tokens are selected, the inputs to transformer blocks 0 to 23 are used for calibration. Our results indicate that, for tuning the rotation matrices in LLaMA3-8B and LLaMA2-7B, using $16 \times 2048$ tokens is often sufficient. We believe that these reasons may all be the causes for the optimization of DFRot being relatively insensitive to the number of tokens in the calibration dataset:

1. These special tokens will appear in relatively shallow network layers (Sun et al., 2024), therefore, a small number of layers are also sufficient to capture these tokens.

2. For a model, tokens with massive activations in LLMs tend to exhibit only a few similar data distributions because these tokens are often produced by out_proj or down_proj layers with large weights (Yu et al., 2024).

3. GPTQ will use 128 samples with a length of 2048 to calibrate the weights, which reduces the impact of the sample size during rotation matrix calibration.

## 4.4 Results on MMLU

| W-A-KV | Methods | LLaMA2-7B | LLaMA3-8B | QWen2-7B | Mistral-7B-v0.3 |
|---|---|---|---|---|---|
| 16-16-16 | FP | 41.85 | 62.23 | 69.47 | 59.11 |
| 4-4-16 | QuaRot | 34.83 | 51.43 | 62.67 | 52.82 |
| 4-4-16 | DFRot | 35.54 | 51.68 | 63.40 | 53.38 |

Table 4: Comparison of MMLU results under different methods.

We compare DFRot with QuaRot with W4A4KV16 quantization configuration with different models. As seen in Table 4, even though rotation matrix $R_1$ is refined with WikiText-2 dataset, DFRot also outperforms QuaRot in all models. It indicates that DFRot, which refines $R_1$ by optimized long tailed quantization error, can be seen as a general method. It is also worth noting that even though DFRot achieves slight improvement with WikiText2 for LLaMA2-7B, it achieves 0.71% improvement with MMLU, which is significant. On the contrary, for the LLaMA3-8B, while DFRot achieves significant improvement with WikiText2, it only achieves 0.25% improvement with MMLU, which is slight. To sum up, we can know that the PPL with WikiText2 can not been seen as a good indicator of the model downstream performance. In the future, we will study how to design more robust quantization algorithms for downstream tasks to further enhance the capabilities of quantized models in downstream tasks.

## 5 Conclusion

Eliminating outliers in LLMs through rotational invariance can significantly improve model quantization accuracy. In this paper, we find that in the context of 4-bit activation quantization, the fundamental reason for the effectiveness difference between RO and RH is their performance on tokens with massive activations. Specifically, randomized Hadamard transformations perform better on these tokens than random Orthogonal transformation. Based on the observation that tokens with massive activations are rare and important in LLMs, we treat the problem as a long-tail optimization and construct a simple yet effective weighted quantization loss function to balance the importance of tokens. Furthermore, by alternately employing orthogonal Procrustes transformations to refine the rotation matrix $R_1$ and optimizing quantization parameters for $X$, our method, named DFRot, enhances the Rotated LLMs by achieving Dual Free, including *Outlier-Free* and *Massive Activation-Free*. It is worth noting that DFRot significantly improves model accuracy in 4-bit activation quantization with just a single data sample, achieving PPL improvements of 0.98 and 0.95 on W4A4KV4 and W4A4KV16, respectively, for the LLaMA3-70B, which is notable for its quantization challenge.

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

# A  Calibration Data

In this section, we explain the reason why we only used a single data sample to calibrate the rotation matrix $R_1$ in DFRot, and don not attempt to use more data:

- In LLMs, outliers and massive activations often appear in some fixed channels. Therefore, the process of optimizing the rotation matrix can be seen as an optimization of the distribution patterns of outliers and massive activations. We have simply use ten samples to calibrate the rotation matrix for LLaMA2-7B, but no significant improvement in accuracy was observed.

- Our calibration data is a sample with a length of 2048 tokens. Since we obtain the calibration set from each MHA and FFN, taking LLaMA2-7B as an example, we can obtain $2048 \times 32 \times 2 = 131072$ tokens as calibration tokens. This is relatively sufficient to statistically analyze the distribution patterns of outliers and massive activations.

# B  Limitations

Due to the discontinuity of the quantization function, our current optimization method is prone to getting stuck in local minima and often relies on initialization. As can be seen from Figure 5 and Figure 6, the performance of RH is better than that of RO in most cases. Meanwhile, during the optimization of the rotation matrix, we also found that after iterative convergence, the final quantization error optimized with RH initialization is often better than that with RO.

# C  Algorithm

---

**Algorithm 1** Optimization of Quantization Parameters and Rotation Matrix

---

**Require:** Token $x$, initial rotation matrix $R_1$, quantization function $\mathcal{Q}$
**Ensure:** Optimized rotation matrix $R_1$ and quantization parameters $\eta_x$
1: Initialize $R_1$ with randomized Hadamard matrix, $t = 0$
2: **while** t≤100 **do**
3:    // Step 1: Optimize Quantization Parameters $\eta_x$
4:    **for** each token $x$ **do**
5:       Compute quantization parameters $s, z$ via $\arg\min_{s,z} \|xR_1^{t-1} - \mathcal{Q}(xR_1^{t-1}, s_x, z_x)\|_2^2$
6:       Update $\eta_x^t = \mathcal{Q}(xR_1^{t-1}, s_x^t, z_x^t)$
7:    **end for**
8:    // Step 2: Optimize Rotation Matrix $R_1$
9:    Solve the Procrustes problem to update $R_1^t$: $R_1^t = \arg\min_R \|XR - \eta_X^t\|_F^2$
10:    $t = t + 1$
11: **end while**
12: **return** Optimized $R_1^*$

---

## D  Quantization error for tokens with Massive activation in LLaMA-7B, LLaMA2-7B, LLaMA2-13B

More quantization results for LLaMA-7B, LLaMA2-7B and LLaMA2-13B:

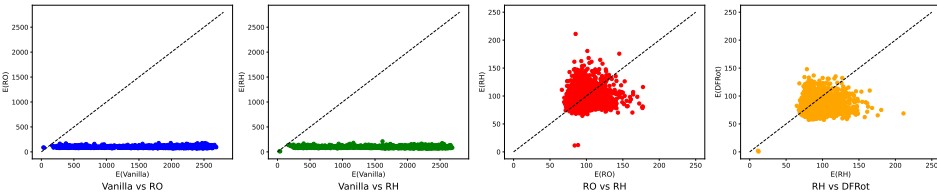

Figure 7: Comparison of 4-bit quantization error for the token with massive activation with NR, RO, RH and DFRot for LLaMA-7B from Figure 2.

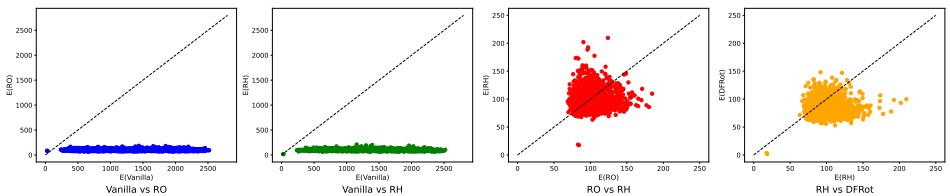

Figure 8: Comparison of 4-bit quantization error for the token with massive activation with NR, RO, RH and DFRot for LLaMA2-7B from Figure 2.

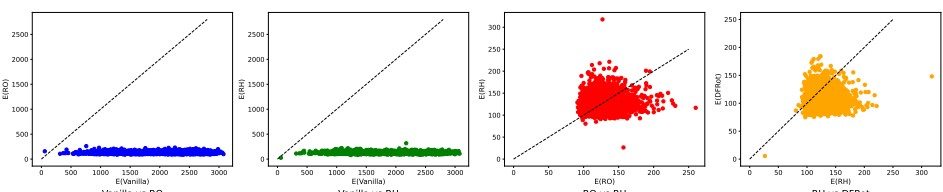

Figure 9: Comparison of 4-bit quantization error for the token with massive activation with NR, RO, RH and DFRot for LLaMA2-13B from Figure 2.

# E  Quantization error between NR, RO, RH and DFRot

More 2D quantization error visualization are shown as follows:

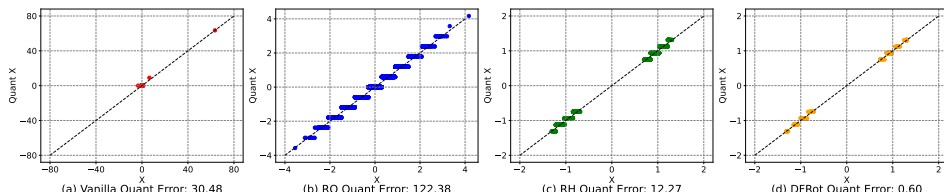

Figure 10: Comparison of 4-bit quantization error for the token with massive activation with NR, RO, RH and DFRot for LLaMA-7B from Figure 2.

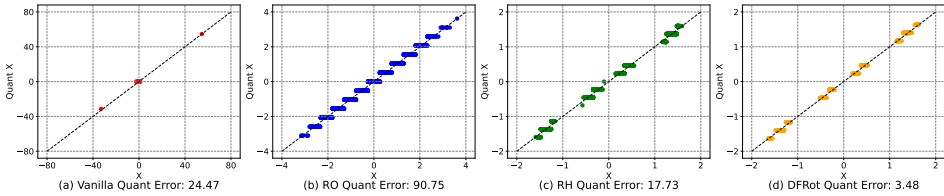

Figure 11: Comparison of 4-bit quantization error for the token with massive activation with NR, RO, RH and DFRot for LLaMA2-7B from Figure 2.

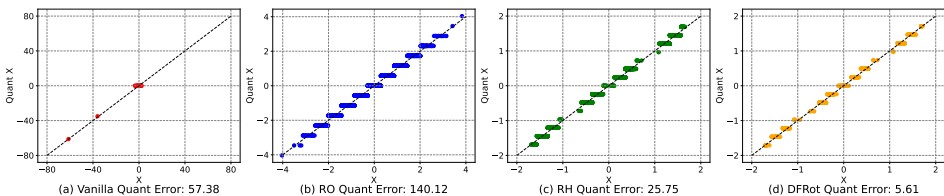

Figure 12: Comparison of 4-bit quantization error for the token with massive activation with NR, RO, RH and DFRot for LLaMA2-13B from Figure 2.

# F    Visualization for Different layers

We visualize for more layer as follwing:

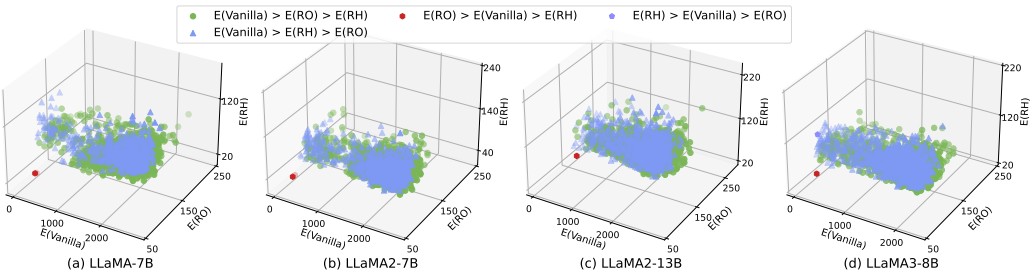

Figure 13: The tokens are from model.layers.4.input_layernorm.

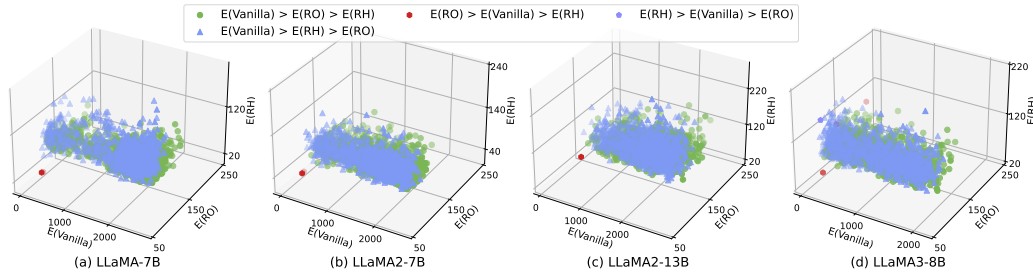

Figure 14: The tokens are from model.layers.4.post_attention_layernorm.

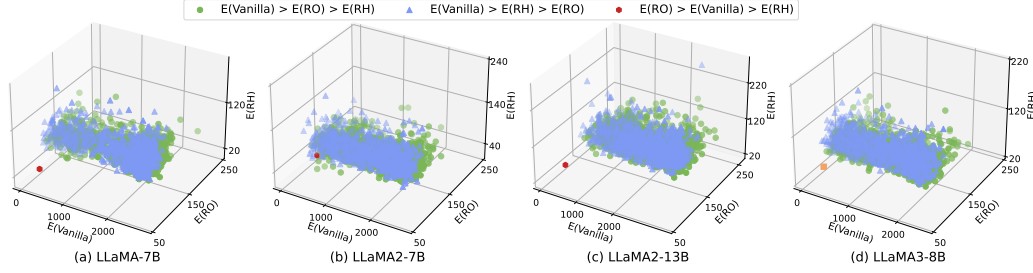

Figure 15: The tokens are from model.layers.8.input_layernorm.

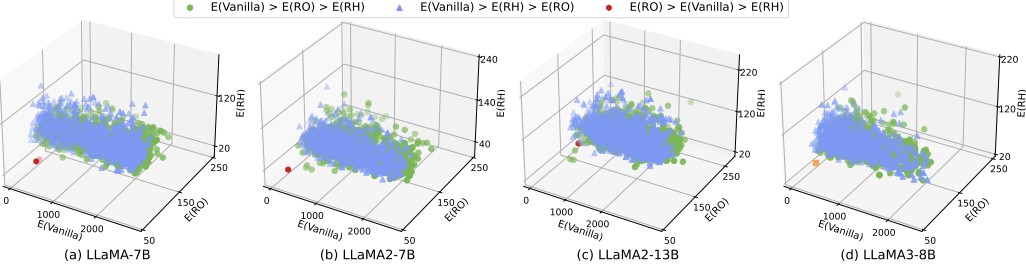

Figure 16: The tokens are from model.layers.8.post_attention_layernorm.

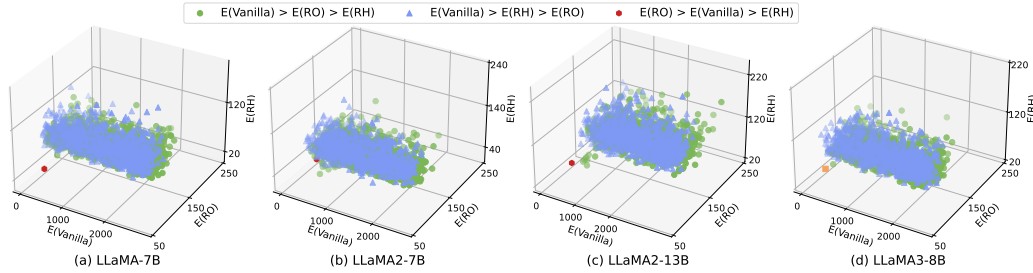

Figure 17: The tokens are from model.layers.12.input_layernorm.

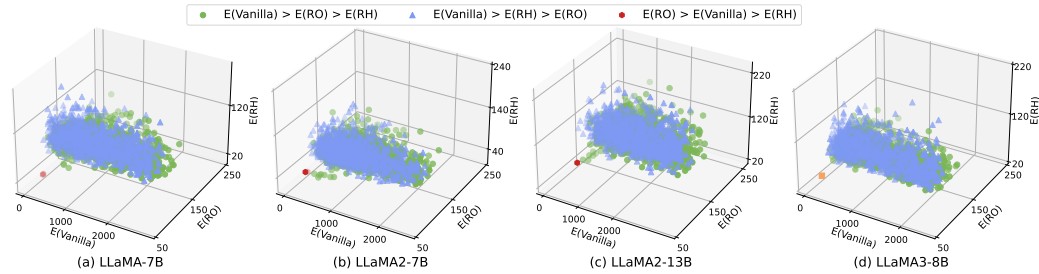

Figure 18: The tokens are from model.layers.12.post_attention_layernorm.

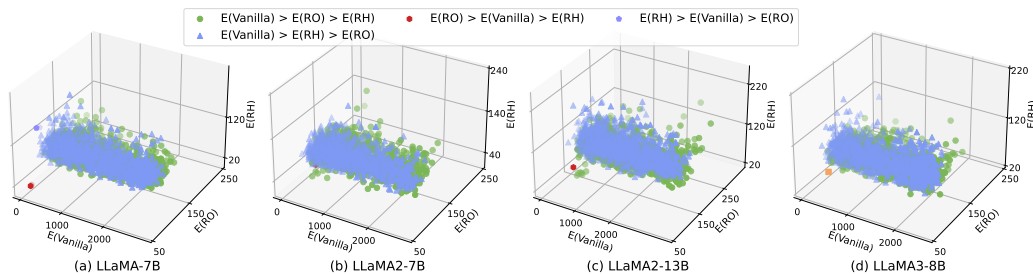

Figure 19: The tokens are from model.layers.16.input_layernorm.

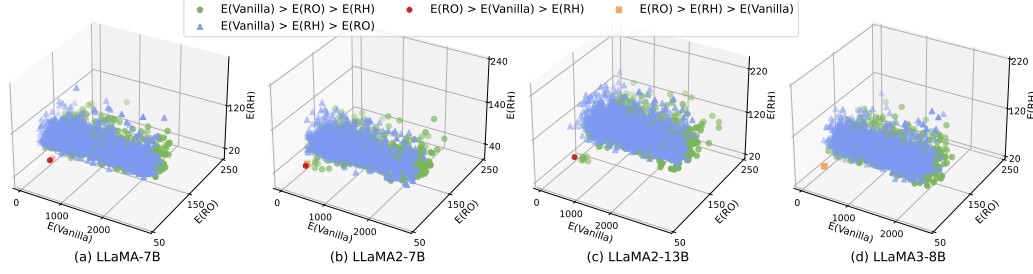

Figure 20: The tokens are from model.layers.16.post_attention_layernorm.

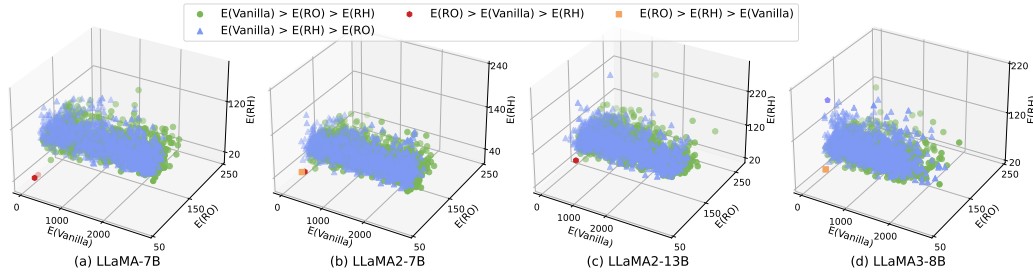

Figure 21: The tokens are from model.layers.20.input_layernorm.

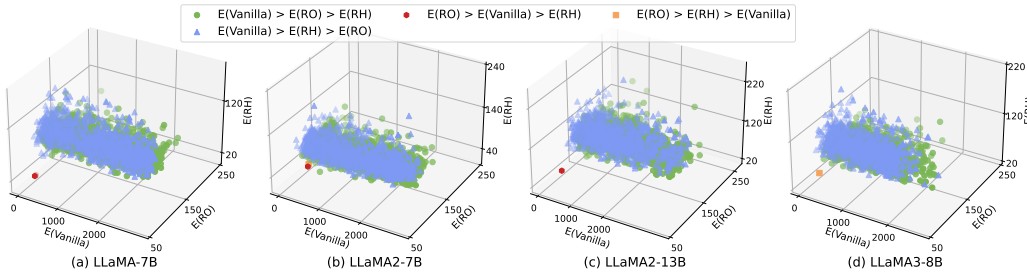

Figure 22: The tokens are from model.layers.20.post_attention_layernorm.

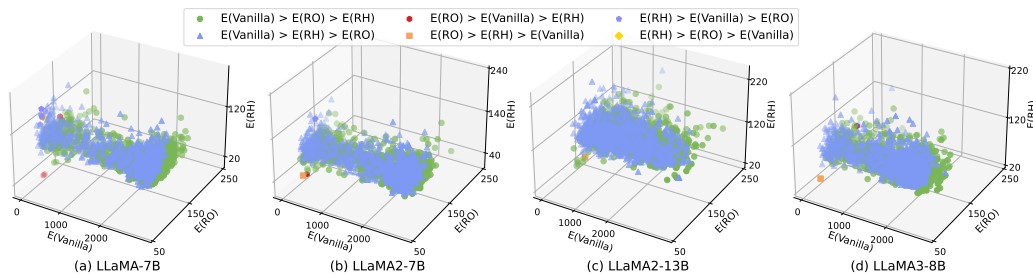

Figure 23: The tokens are from model.layers.24.input_layernorm.

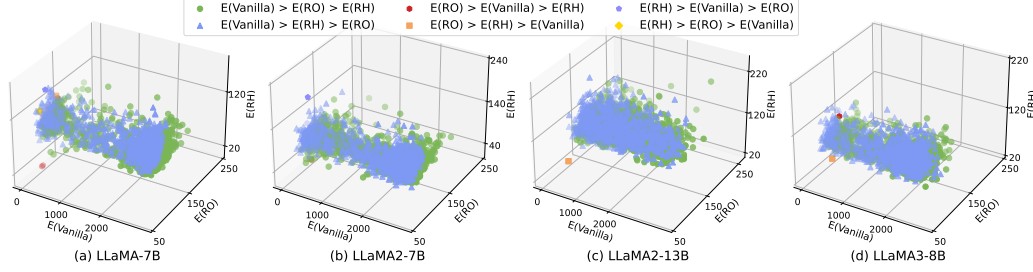

Figure 24: The tokens are from model.layers.24.post_attention_layernorm.

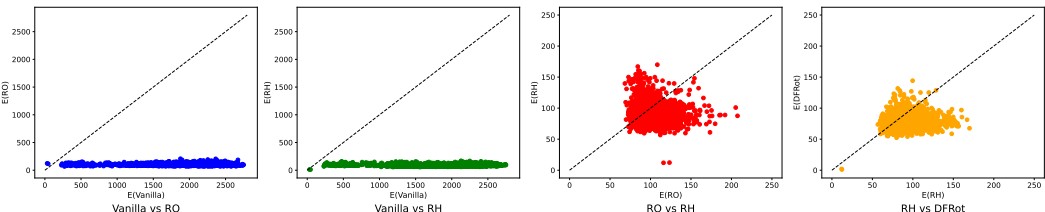

Figure 25: Comparison of 4-bit quantization error for the token with massive activation with NR, RO, RH and DFRot for LLaMA-7B from Figure 13.

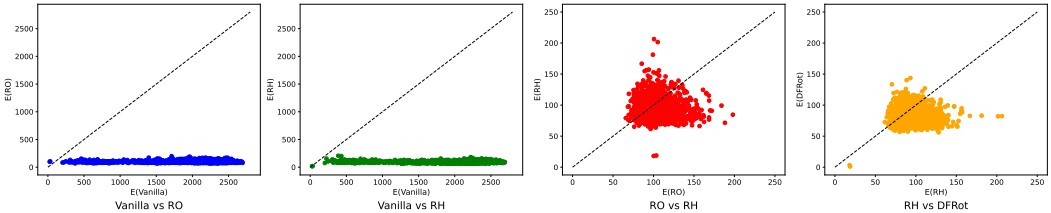

Figure 26: Comparison of 4-bit quantization error for the token with massive activation with NR, RO, RH and DFRot for LLaMA2-7B from Figure 13.

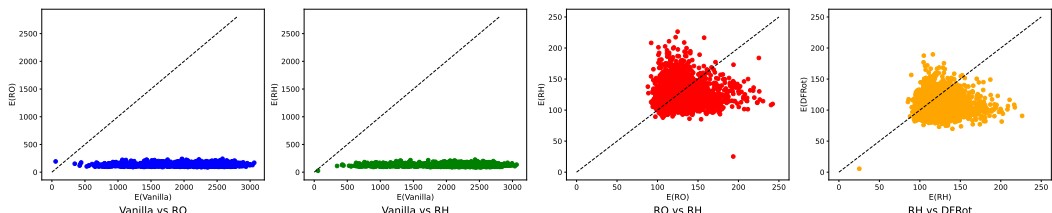

Figure 27: Comparison of 4-bit quantization error for the token with massive activation with NR, RO, RH and DFRot for LLaMA2-13B from Figure 13.

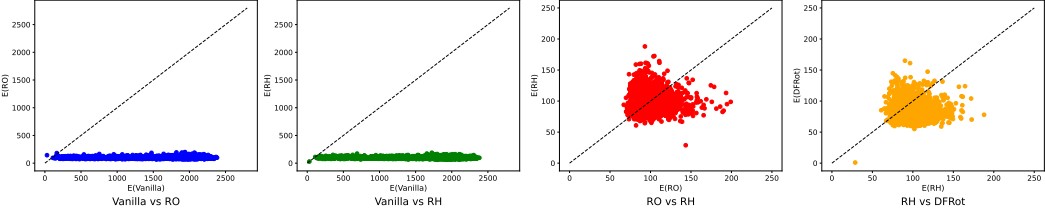

Figure 28: Comparison of 4-bit quantization error for the token with massive activation with NR, RO, RH and DFRot for LLaMA3-8B from Figure 13.

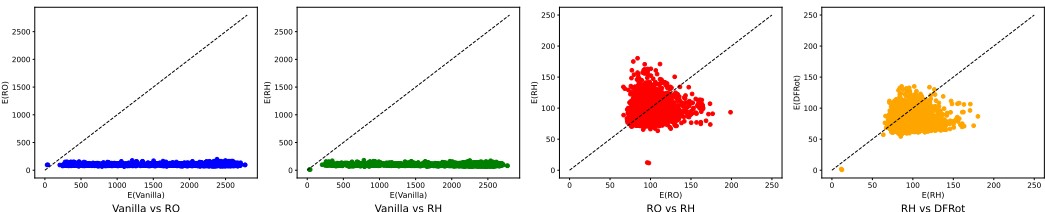

Figure 29: Comparison of 4-bit quantization error for the token with massive activation with NR, RO, RH and DFRot for LLaMA-7B from Figure 15.

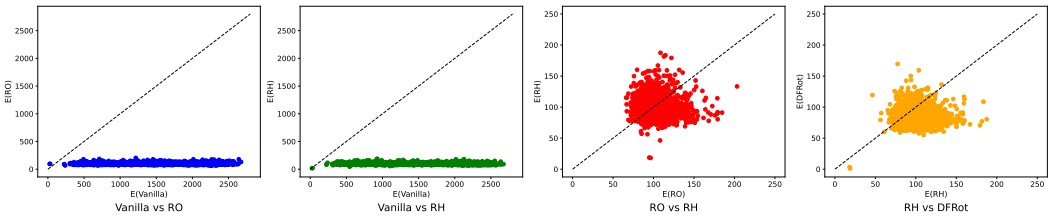

Figure 30: Comparison of 4-bit quantization error for the token with massive activation with NR, RO, RH and DFRot for LLaMA2-7B from Figure 15.

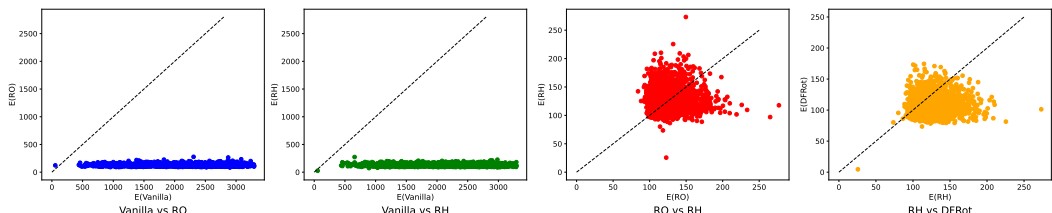

Figure 31: Comparison of 4-bit quantization error for the token with massive activation with NR, RO, RH and DFRot for LLaMA2-13B from Figure 15.

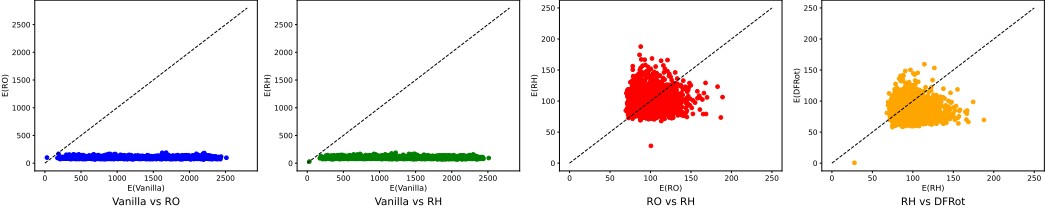

Figure 32: Comparison of 4-bit quantization error for the token with massive activation with NR, RO, RH and DFRot for LLaMA3-8B from Figure 15.

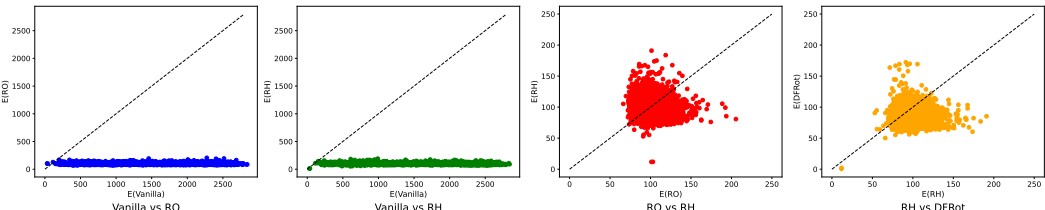

Figure 33: Comparison of 4-bit quantization error for the token with massive activation with NR, RO, RH and DFRot for LLaMA-7B from Figure 17.

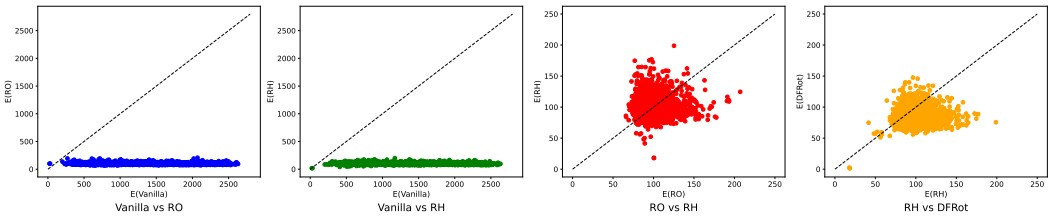

Figure 34: Comparison of 4-bit quantization error for the token with massive activation with NR, RO, RH and DFRot for LLaMA2-7B from Figure 17.

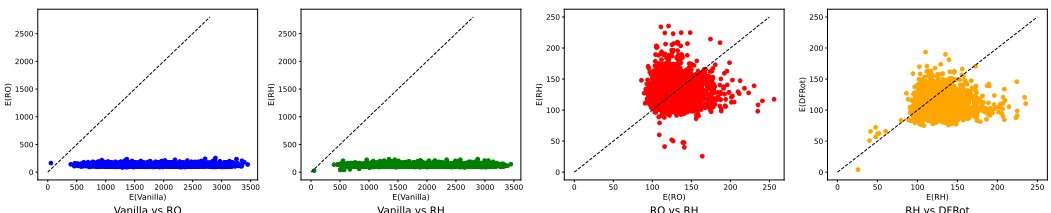

Figure 35: Comparison of 4-bit quantization error for the token with massive activation with NR, RO, RH and DFRot for LLaMA2-13B from Figure 17.

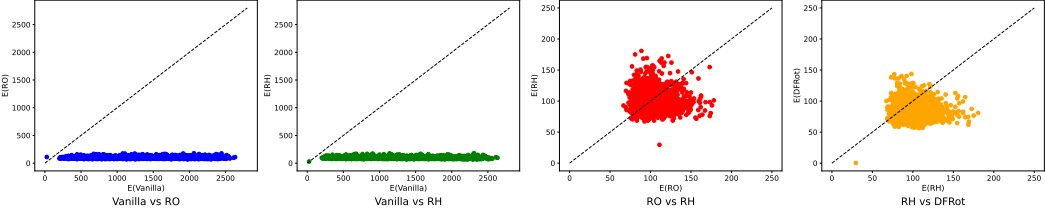

Figure 36: Comparison of 4-bit quantization error for the token with massive activation with NR, RO, RH and DFRot for LLaMA3-8B from Figure 17.

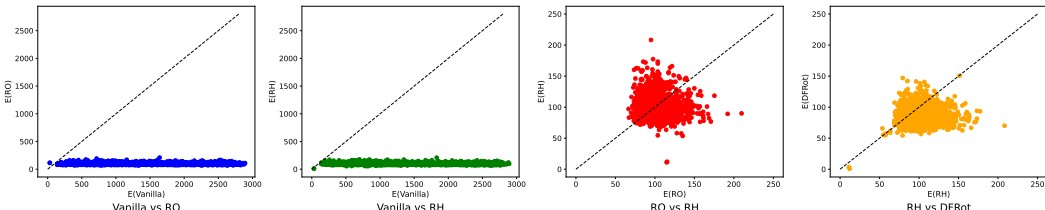

Figure 37: Comparison of 4-bit quantization error for the token with massive activation with NR, RO, RH and DFRot for LLaMA-7B from Figure 19.

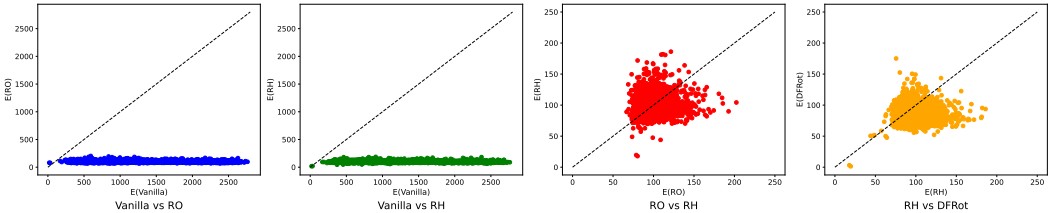

Figure 38: Comparison of 4-bit quantization error for the token with massive activation with NR, RO, RH and DFRot for LLaMA2-7B from Figure 19.

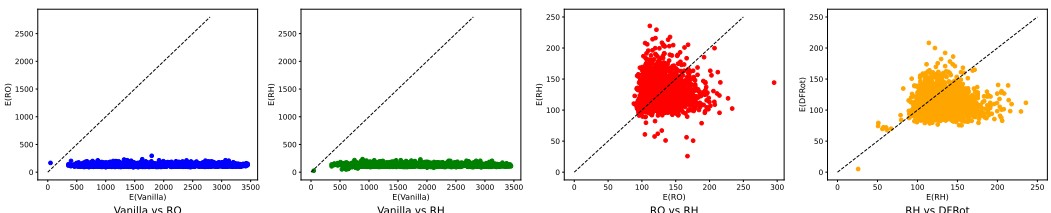

Figure 39: Comparison of 4-bit quantization error for the token with massive activation with NR, RO, RH and DFRot for LLaMA2-13B from Figure 19.

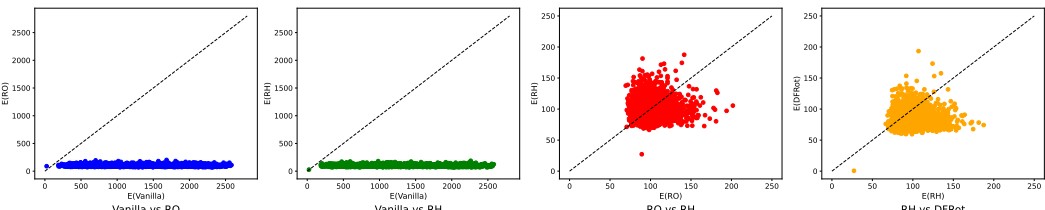

Figure 40: Comparison of 4-bit quantization error for the token with massive activation with NR, RO, RH and DFRot for LLaMA3-8B from Figure 19.

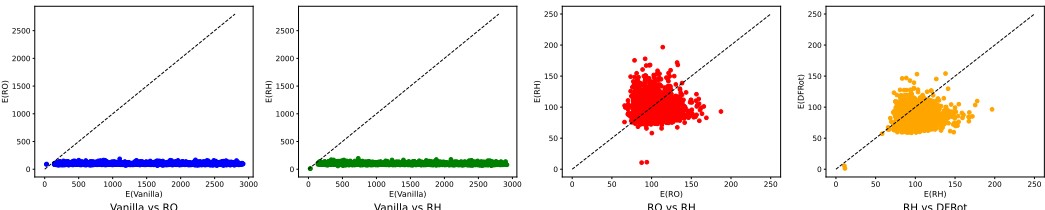

Figure 41: Comparison of 4-bit quantization error for the token with massive activation with NR, RO, RH and DFRot for LLaMA-7B from Figure 21.

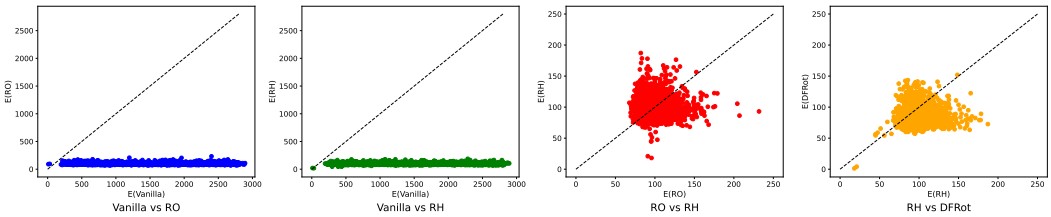

Figure 42: Comparison of 4-bit quantization error for the token with massive activation with NR, RO, RH and DFRot for LLaMA2-7B from Figure 21.

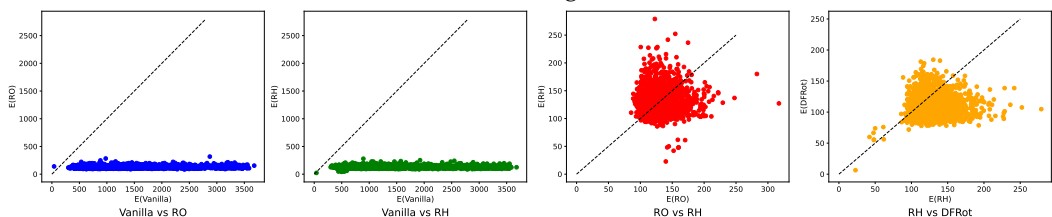

Figure 43: Comparison of 4-bit quantization error for the token with massive activation with NR, RO, RH and DFRot for LLaMA2-13B from Figure 21.

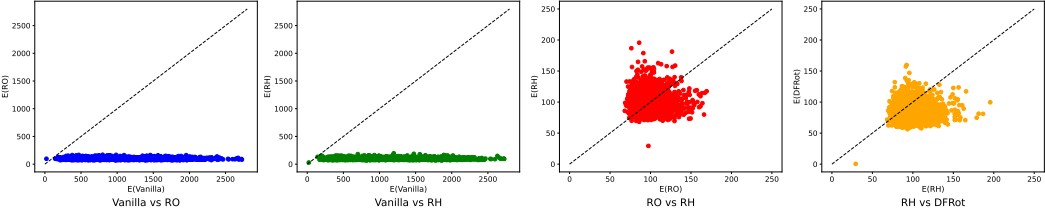

Figure 44: Comparison of 4-bit quantization error for the token with massive activation with NR, RO, RH and DFRot for LLaMA3-8B from Figure 21.

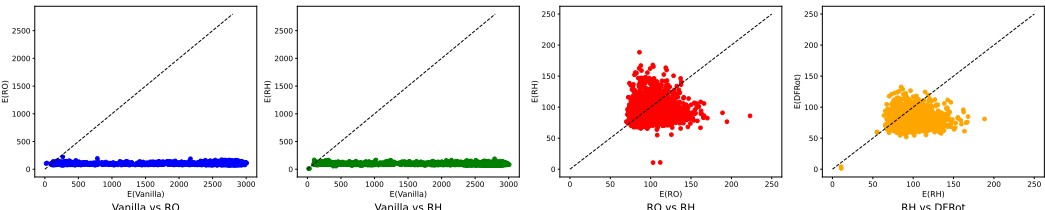

Figure 45: Comparison of 4-bit quantization error for the token with massive activation with NR, RO, RH and DFRot for LLaMA-7B from Figure 23.

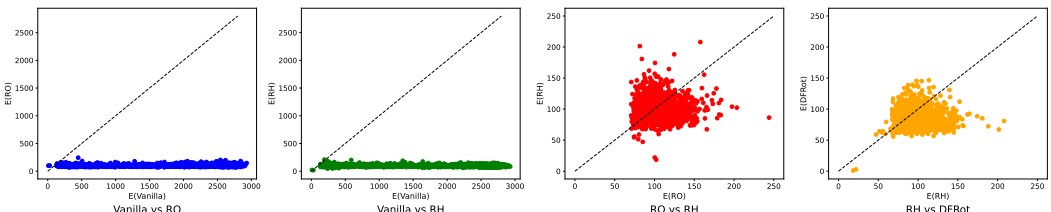

Figure 46: Comparison of 4-bit quantization error for the token with massive activation with NR, RO, RH and DFRot for LLaMA2-7B from Figure 23.

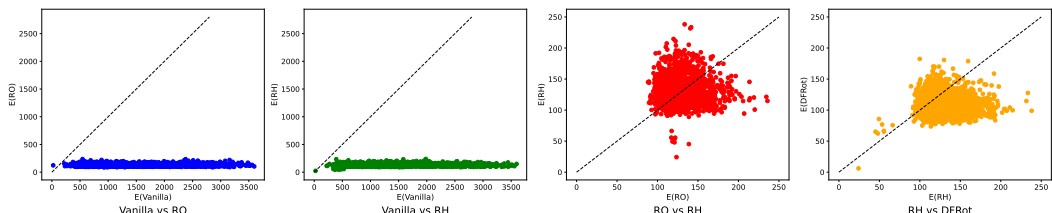

Figure 47: Comparison of 4-bit quantization error for the token with massive activation with NR, RO, RH and DFRot for LLaMA2-13B from Figure 23.

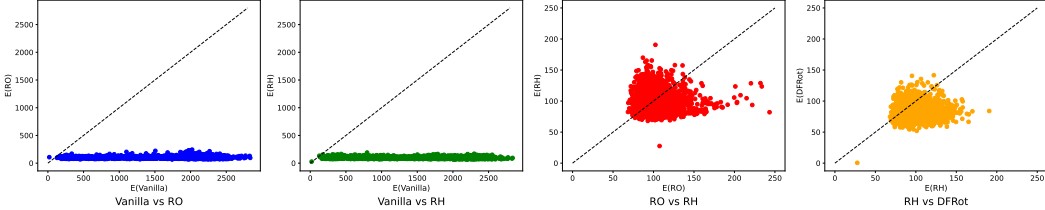

Figure 48: Comparison of 4-bit quantization error for the token with massive activation with NR, RO, RH and DFRot for LLaMA3-8B from Figure 23.

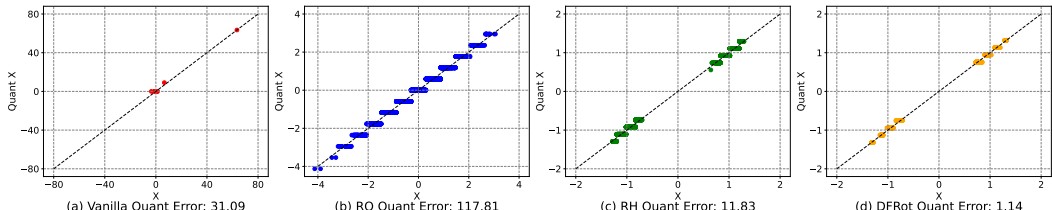

Figure 49: Comparison of 4-bit quantization error for the token with massive activation with NR, RO, RH and DFRot for LLaMA-7B from Figure 13.

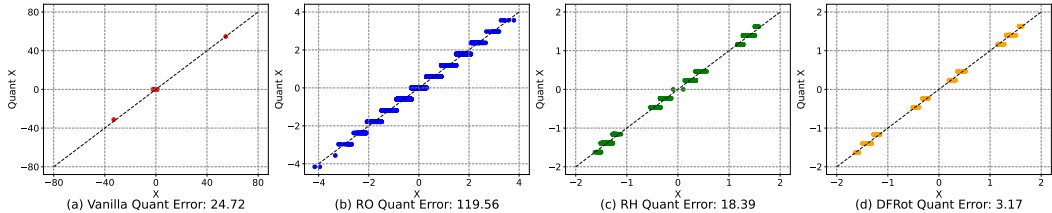

Figure 50: Comparison of 4-bit quantization error for the token with massive activation with NR, RO, RH and DFRot for LLaMA2-7B from Figure 13.

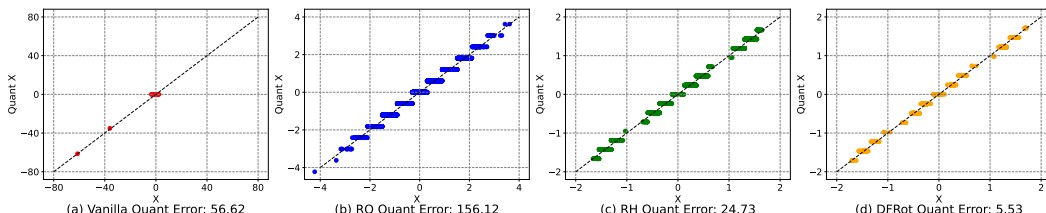

Figure 51: Comparison of 4-bit quantization error for the token with massive activation with NR, RO, RH and DFRot for LLaMA2-13B from Figure 13.

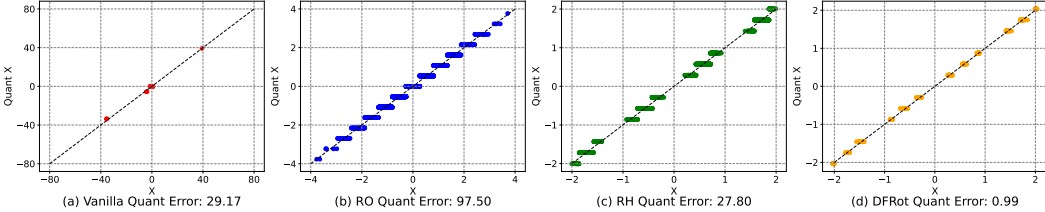

Figure 52: Comparison of 4-bit quantization error for the token with massive activation with NR, RO, RH and DFRot for LLaMA3-8B from Figure 13.

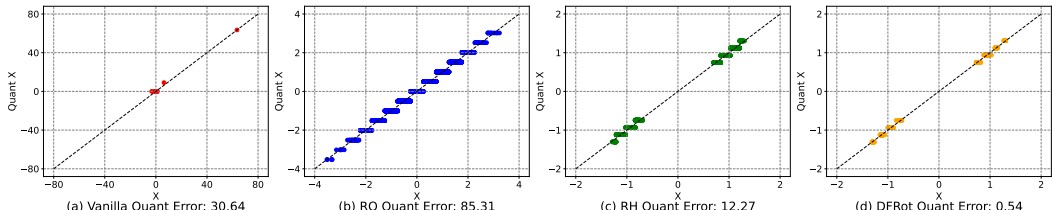

Figure 53: Comparison of 4-bit quantization error for the token with massive activation with NR, RO, RH and DFRot for LLaMA-7B from Figure 15.

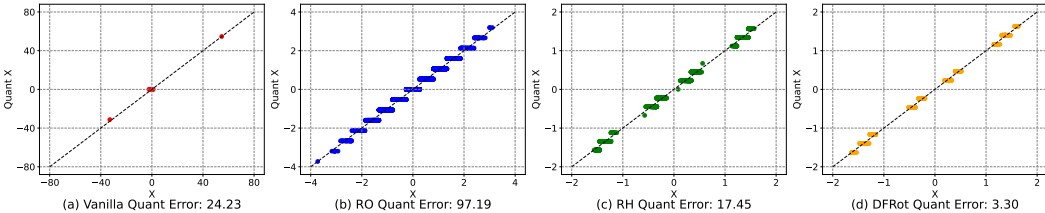

Figure 54: Comparison of 4-bit quantization error for the token with massive activation with NR, RO, RH and DFRot for LLaMA2-7B from Figure 15.

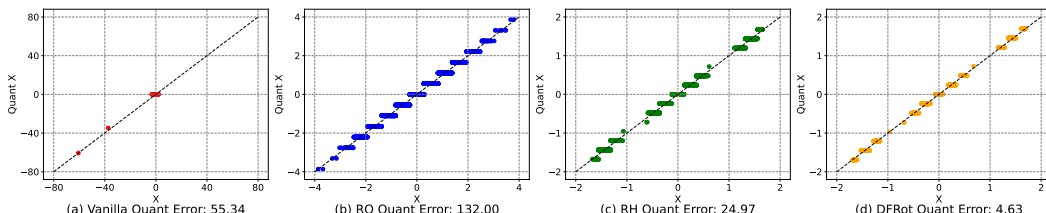

Figure 55: Comparison of 4-bit quantization error for the token with massive activation with NR, RO, RH and DFRot for LLaMA2-13B from Figure 15.

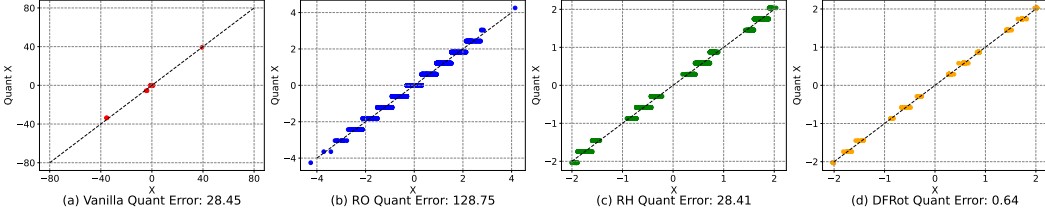

Figure 56: Comparison of 4-bit quantization error for the token with massive activation with NR, RO, RH and DFRot for LLaMA3-8B from Figure 15.

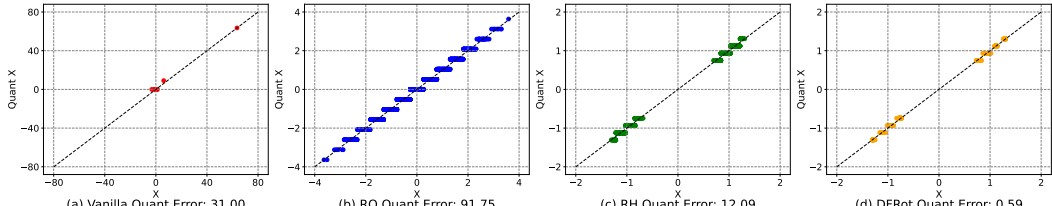

Figure 57: Comparison of 4-bit quantization error for the token with massive activation with NR, RO, RH and DFRot for LLaMA-7B from Figure 17.

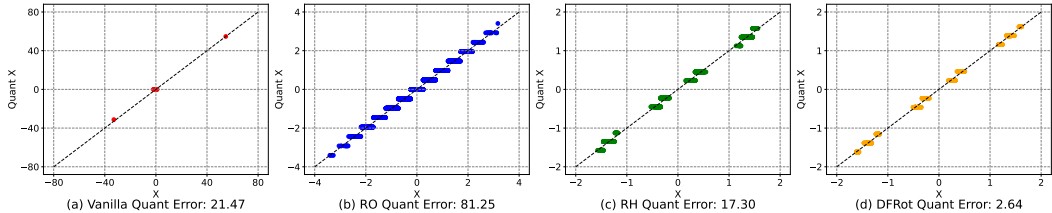

Figure 58: Comparison of 4-bit quantization error for the token with massive activation with NR, RO, RH and DFRot for LLaMA2-7B from Figure 17.

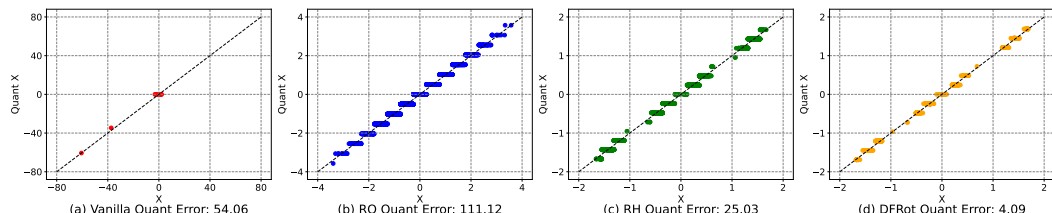

Figure 59: Comparison of 4-bit quantization error for the token with massive activation with NR, RO, RH and DFRot for LLaMA2-13B from Figure 17.

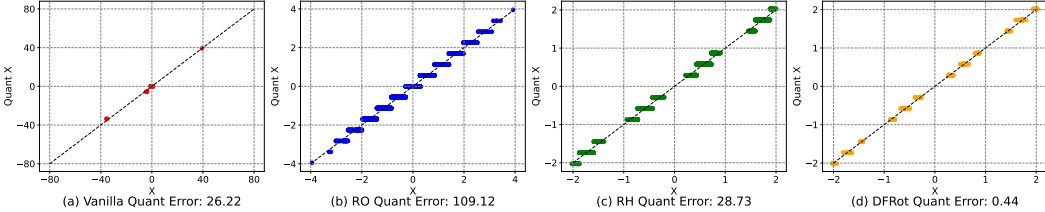

Figure 60: Comparison of 4-bit quantization error for the token with massive activation with NR, RO, RH and DFRot for LLaMA3-8B from Figure 17.

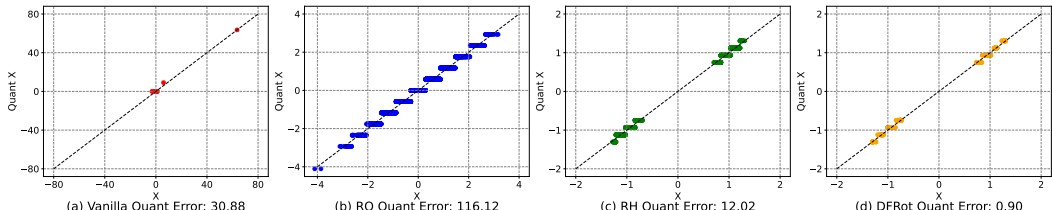

Figure 61: Comparison of 4-bit quantization error for the token with massive activation with NR, RO, RH and DFRot for LLaMA-7B from Figure 19.

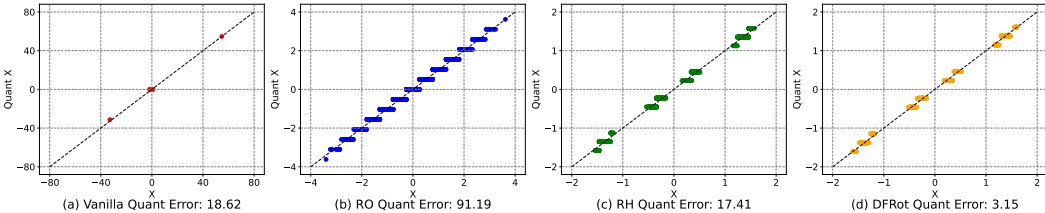

Figure 62: Comparison of 4-bit quantization error for the token with massive activation with NR, RO, RH and DFRot for LLaMA2-7B from Figure 19.

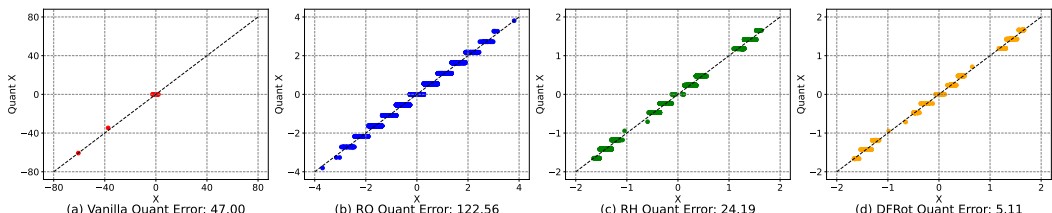

Figure 63: Comparison of 4-bit quantization error for the token with massive activation with NR, RO, RH and DFRot for LLaMA2-13B from Figure 19.

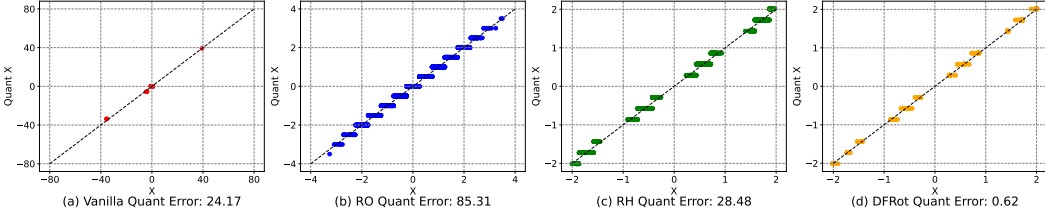

Figure 64: Comparison of 4-bit quantization error for the token with massive activation with NR, RO, RH and DFRot for LLaMA3-8B from Figure 19.

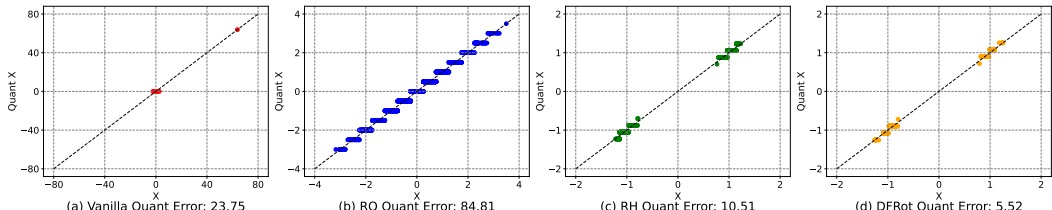

Figure 65: Comparison of 4-bit quantization error for the token with massive activation with NR, RO, RH and DFRot for LLaMA-7B from Figure 21.

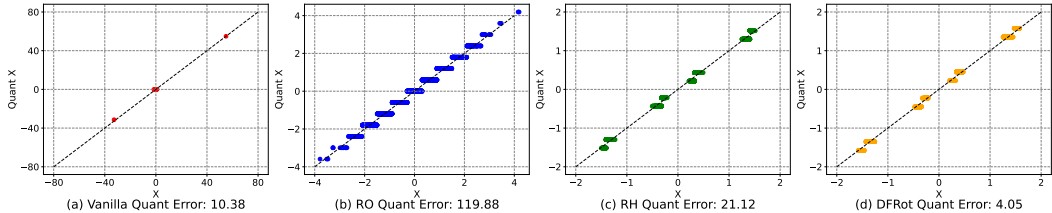

Figure 66: Comparison of 4-bit quantization error for the token with massive activation with NR, RO, RH and DFRot for LLaMA2-7B from Figure 21.

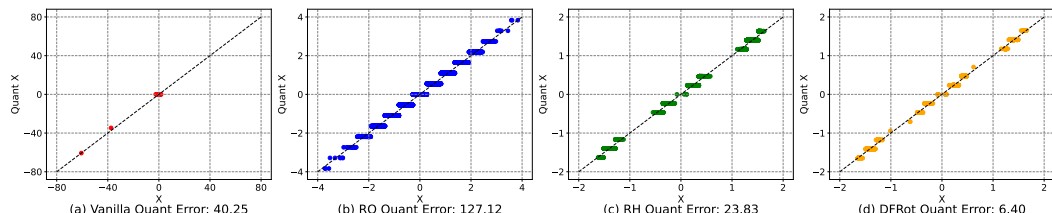

Figure 67: Comparison of 4-bit quantization error for the token with massive activation with NR, RO, RH and DFRot for LLaMA2-13B from Figure 21.

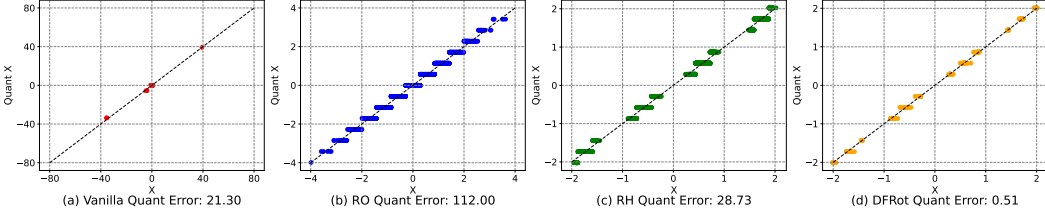

Figure 68: Comparison of 4-bit quantization error for the token with massive activation with NR, RO, RH and DFRot for LLaMA3-8B from Figure 21.

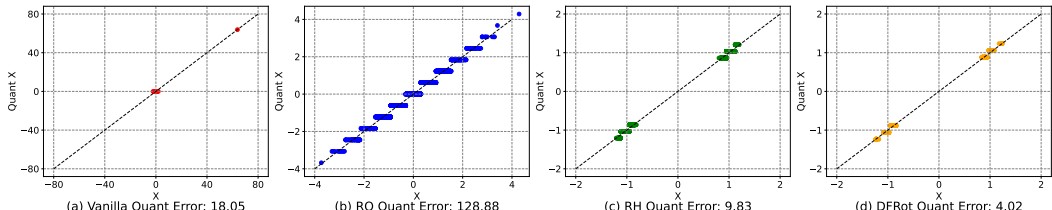

Figure 69: Comparison of 4-bit quantization error for the token with massive activation with NR, RO, RH and DFRot for LLaMA-7B from Figure 23.

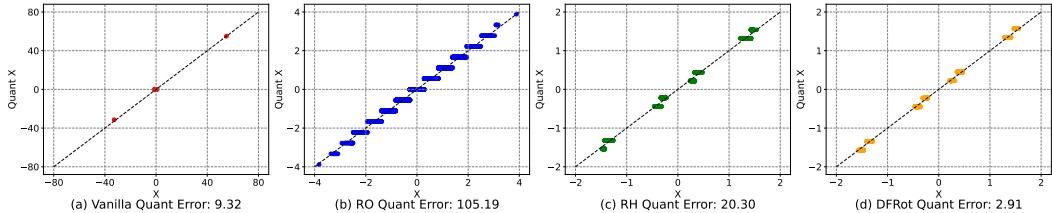

Figure 70: Comparison of 4-bit quantization error for the token with massive activation with NR, RO, RH and DFRot for LLaMA2-7B from Figure 23.

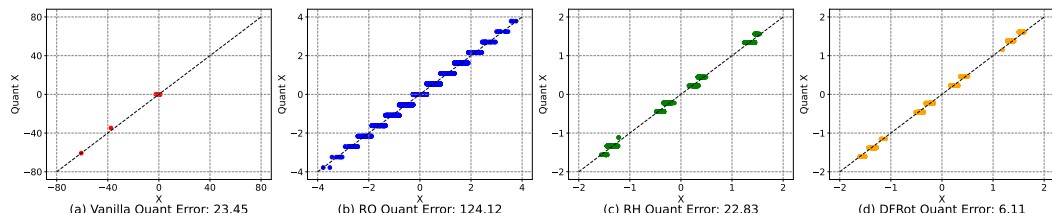

Figure 71: Comparison of 4-bit quantization error for the token with massive activation with NR, RO, RH and DFRot for LLaMA2-13B from Figure 23.

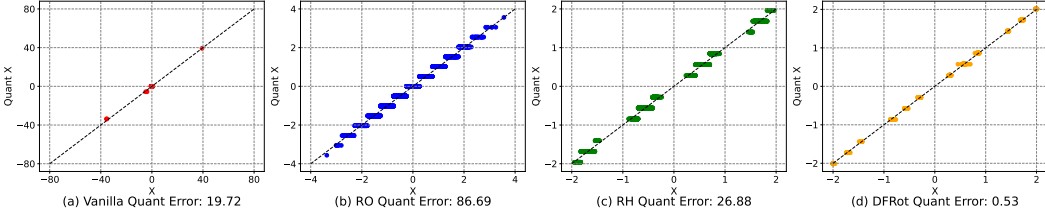

Figure 72: Comparison of 4-bit quantization error for the token with massive activation with NR, RO, RH and DFRot for LLaMA3-8B from Figure 23.

# G Full Results

| Model | #Bits W-A-KV | Method | ARC-c (↑) | ARC-e (↑) | BoolQ (↑) | HellaS. (↑) | Lam. (↑) | OBQA (↑) | PIQA (↑) | SIQA (↑) | WinoG. (↑) | Avg. (↑) | Wiki2 (↓) |
|---|---|---|---|---|---|---|---|---|---|---|---|---|---|
| 2-7B | 16-16-16 | Full Precision | 46.42 | 74.33 | 77.71 | 75.94 | 73.69 | 44.20 | 79.16 | 45.91 | 69.53 | 65.21 | 5.47 |
| | 4-4-16 | RTN | 25.34 | 28.03 | 50.52 | 27.71 | 1.01 | 26.20 | 50.82 | 33.93 | 48.38 | 32.44 | nan |
| | | SmoothQuant | 28.33 | 26.39 | 49.39 | 27.28 | 1.18 | 23.40 | 48.80 | 33.62 | 50.75 | 32.13 | nan |
| | | GPTQ | 24.40 | 28.70 | 51.62 | 28.66 | 1.36 | 24.60 | 51.14 | 34.49 | 49.49 | 32.72 | nan |
| | | QuaRot | 45.99 | 72.73 | 77.8 | 75.92 | 73.45 | 43.8 | 78.02 | 46.16 | 68.03 | 64.66 | 5.45 |
| | | **DFRot** | 44.97 | 70.24 | 74.34 | 73.23 | 68.99 | 42.60 | 76.82 | 44.06 | 66.54 | 62.42 | 6.13 |
| | | SpinQuant* | 37.54 | 62.58 | 71.16 | 70.48 | 67.16 | 34.80 | 75.46 | 39.76 | 60.62 | 57.37 | 6.78 |
| | | OSTQuant* | 44.03 | 71.93 | 75.41 | 74.94 | 73.22 | 43.20 | 78.51 | 45.85 | 68.03 | 63.90 | 5.60 |
| | 4-4-4 | RTN | 27.22 | 27.06 | 50.83 | 27.34 | 0.93 | 25.80 | 49.51 | 34.85 | 50.51 | 32.67 | nan |
| | | SmoothQuant | 26.37 | 25.63 | 47.71 | 27.05 | 1.11 | 26.40 | 51.90 | 34.49 | 48.38 | 32.12 | nan |
| | | GPTQ | 26.96 | 27.65 | 52.84 | 28.83 | 1.63 | 29.20 | 49.62 | 35.11 | 33.52 | nan | |
| | | Omniquant | 31.40 | 53.75 | 63.79 | 55.06 | 35.63 | 34.40 | 66.59 | 40.28 | 54.70 | 48.40 | 14.26 |
| | | QuaRot | 41.3 | 69.32 | 72.66 | 72.09 | 69.67 | 39.0 | 76.5 | 43.19 | 63.54 | 60.81 | 6.25 |
| | | **DFRot** | 43.52 | 70.83 | 73.30 | 72.62 | 68.46 | 41.40 | 76.82 | 44.17 | 65.11 | 61.80 | 6.25 |
| | | SpinQuant* | 40.44 | 71.08 | 74.40 | 73.51 | 70.66 | 41.80 | 76.88 | 43.50 | 65.82 | 62.01 | 5.96 |
| | | OSTQuant* | 42.92 | 72.56 | 74.71 | 73.14 | 71.76 | 44.40 | 77.42 | 44.98 | 66.77 | 63.18 | 5.91 |
| 2-13B | 16-16-16 | Full Precision | 49.15 | 77.53 | 80.58 | 79.39 | 76.62 | 45.20 | 80.63 | 47.49 | 71.90 | 67.61 | 4.88 |
| | 4-4-16 | RTN | 27.99 | 26.81 | 38.50 | 26.08 | 0.00 | 23.60 | 48.20 | 34.90 | 51.62 | 30.86 | 8e3 |
| | | SmoothQuant | 24.49 | 35.06 | 47.98 | 30.87 | 3.67 | 26.20 | 55.01 | 35.31 | 49.72 | 34.26 | 1e3 |
| | | GPTQ | 27.82 | 26.77 | 37.92 | 25.67 | 0.00 | 21.80 | 47.77 | 35.11 | 48.15 | 30.11 | 4e3 |
| | | QuaRot | 46.42 | 73.86 | 78.10 | 75.68 | 74.31 | 43.00 | 79.05 | 44.37 | 71.35 | 65.13 | 5.35 |
| | | **DFRot** | 46.67 | 74.45 | 77.19 | 77.07 | 75.04 | 43.6 | 78.29 | 46.16 | 69.61 | 65.34 | 5.39 |
| | | SpinQuant* | 43.77 | 69.99 | 76.57 | 74.63 | 72.81 | 41.60 | 77.20 | 44.27 | 68.19 | 63.23 | 5.24 |
| | | OSTQuant* | 47.78 | 74.66 | 80.03 | 77.60 | 75.94 | 44.40 | 79.38 | 46.06 | 70.32 | 66.24 | 5.14 |
| | 4-4-4 | RTN | 27.82 | 26.52 | 38.38 | 26.27 | 0.02 | 26.00 | 49.78 | 34.39 | 49.17 | 30.93 | 7e3 |
| | | SmoothQuant | 24.49 | 33.00 | 45.84 | 30.70 | 2.70 | 23.80 | 53.81 | 34.80 | 51.07 | 33.36 | 2e3 |
| | | GPTQ | 27.90 | 26.39 | 37.95 | 26.16 | 0.00 | 27.00 | 48.26 | 34.39 | 50.43 | 27.85 | 5e3 |
| | | Omniquant | 32.85 | 55.13 | 64.34 | 60.13 | 42.85 | 33.40 | 68.17 | 39.76 | 56.51 | 50.35 | 12.30 |
| | | QuaRot | 44.37 | 73.32 | 77.58 | 75.73 | 73.16 | 43.0 | 78.84 | 45.04 | 69.30 | 64.44 | 5.49 |
| | | **DFRot** | 46.50 | 73.48 | 76.67 | 76.83 | 73.92 | 43.00 | 79.27 | 45.55 | 69.30 | 64.95 | 5.43 |
| | | SpinQuant* | 46.67 | 74.49 | 76.76 | 75.22 | 72.19 | 42.40 | 78.29 | 43.45 | 67.72 | 64.13 | 5.74 |
| | | OSTQuant* | 47.10 | 75.21 | 77.46 | 76.71 | 75.14 | 44.60 | 78.67 | 45.75 | 68.03 | 65.41 | 5.25 |
| 3-8B | 16-16-16 | Full Precision | 53.50 | 77.74 | 81.10 | 79.18 | 75.74 | 44.80 | 80.63 | 47.08 | 73.01 | 68.09 | 6.14 |
| | 4-4-16 | RTN | 23.72 | 30.89 | 46.30 | 31.26 | 3.03 | 27.60 | 52.72 | 35.26 | 50.04 | 33.42 | 6e2 |
| | | SmoothQuant | 23.29 | 28.28 | 48.93 | 29.19 | 1.57 | 28.60 | 54.46 | 33.37 | 49.64 | 33.04 | 1e3 |
| | | GPTQ | 23.46 | 32.07 | 43.79 | 30.10 | 2.41 | 28.00 | 53.97 | 34.14 | 48.86 | 32.98 | 6e2 |
| | | QuaRot | 44.03 | 69.74 | 71.90 | 73.16 | 67.26 | 42.4 | 76.71 | 45.04 | 66.46 | 61.86 | 8.11 |
| | | **DFRot** | 46.16 | 72.90 | 73.73 | 74.98 | 68.23 | 40.60 | 77.53 | 44.32 | 68.67 | 63.01 | 7.78 |
| | | SpinQuant* | 47.35 | 74.12 | 76.36 | 75.98 | 69.88 | 42.46 | 77.37 | 44.47 | 68.98 | 64.11 | 7.28 |
| | | OSTQuant* | 48.81 | 73.48 | 79.82 | 75.97 | 72.62 | 42.40 | 78.18 | 45.75 | 69.22 | 65.14 | 7.24 |
| | 4-4-4 | RTN | 23.72 | 30.56 | 46.18 | 29.83 | 2.70 | 28.60 | 52.45 | 34.39 | 50.20 | 33.18 | 7e2 |
| | | SmoothQuant | 23.55 | 28.96 | 48.84 | 28.90 | 1.44 | 29.40 | 51.09 | 34.14 | 50.36 | 32.96 | 1e3 |
| | | GPTQ | 23.38 | 32.74 | 44.34 | 29.72 | 2.39 | 29.80 | 54.95 | 34.75 | 51.30 | 33.71 | 6e2 |
| | | Omniquant | 22.87 | 30.35 | 41.53 | 31.11 | 1.86 | 25.40 | 53.37 | 34.08 | 50.43 | 32.33 | 4e2 |
| | | QuaRot | 43.17 | 70.58 | 72.66 | 72.53 | 66.66 | 39.20 | 76.06 | 44.83 | 66.77 | 61.38 | 8.28 |
| | | **DFRot** | 44.97 | 71.09 | 73.27 | 74.13 | 67.63 | 43.00 | 78.24 | 44.58 | 69.53 | 62.94 | 7.91 |
| | | SpinQuant* | 46.33 | 73.57 | 76.15 | 75.43 | 71.40 | 41.40 | 79.16 | 44.68 | 68.75 | 64.10 | 7.35 |
| | | OSTQuant* | 49.32 | 76.73 | 78.87 | 76.01 | 70.77 | 43.20 | 78.51 | 45.70 | 69.22 | 65.37 | 7.29 |

Table 5: Complete comparison of the perplexity score on WikiText2 and averaged accuracy on Zero-shot Common Sense Reasoning tasks on **LLaMA-2 & 3**.

| Model | #Bits W-A-KV | Method | ARC-c (↑) | ARC-e (↑) | BoolQ (↑) | HellaS. (↑) | LambA. (↑) | OBQA (↑) | PIQA (↑) | SIQA (↑) | WinoG. (↑) | Avg. (↑) | Wiki2 (↓) |
|---|---|---|---|---|---|---|---|---|---|---|---|---|---|
| 7B | 16-16-16 | Full Precision | 44.71 | 72.90 | 74.98 | 76.20 | 73.08 | 43.80 | 79.16 | 45.55 | 69.93 | 64.48 | 5.68 |
| | 4-4-16 | RTN | 23.46 | 29.34 | 45.05 | 29.02 | 1.24 | 26.00 | 52.07 | 35.11 | 51.30 | 32.51 | 7e3 |
| | | SmoothQuant | 25.17 | 31.40 | 51.62 | 29.73 | 5.43 | 28.20 | 54.68 | 34.44 | 49.09 | 34.42 | 3e2 |
| | | GPTQ | 23.89 | 27.74 | 42.87 | 28.49 | 1.28 | 27.40 | 51.00 | 36.23 | 50.20 | 32.12 | 1e3 |
| | | QuaRot | 41.38 | 68.01 | 72.23 | 73.02 | 70.33 | 42.6 | 76.99 | 43.76 | 66.54 | 61.65 | 6.33 |
| | | **DFRot** | 42.06 | 69.65 | 72.84 | 73.19 | 70.54 | 41.60 | 77.09 | 45.14 | 68.11 | 62.25 | 6.30 |
| | | SpinQuant* | 40.19 | 68.43 | 72.35 | 72.91 | 70.68 | 41.20 | 77.75 | 44.17 | 68.67 | 61.82 | 6.08 |
| | | OSTQuant* | 42.58 | 70.79 | 72.87 | 74.06 | 70.77 | 43.40 | 77.69 | 45.04 | 67.25 | 62.72 | 6.04 |
| | 4-4-4 | RTN | 23.89 | 29.59 | 46.67 | 28.37 | 1.13 | 26.40 | 52.99 | 35.21 | 51.54 | 32.87 | 1e4 |
| | | SmoothQuant | 23.38 | 30.18 | 50.03 | 29.67 | 4.89 | 24.60 | 51.74 | 34.75 | 50.67 | 33.32 | 3e2 |
| | | GPTQ | 23.89 | 27.90 | 43.88 | 27.86 | 1.05 | 26.20 | 51.85 | 34.08 | 49.49 | 31.80 | 2e3 |
| | | Omniquant | 31.40 | 54.84 | 61.80 | 56.98 | 38.29 | 31.80 | 66.59 | 39.30 | 55.17 | 48.46 | 11.26 |
| | | QuaRot | 40.70 | 68.39 | 71.96 | 72.46 | 70.37 | 40.60 | 77.09 | 43.60 | 65.75 | 61.21 | 6.36 |
| | | **DFRot** | 40.87 | 69.65 | 73.24 | 72.96 | 70.13 | 42.40 | 77.04 | 43.86 | 66.38 | 61.84 | 6.36 |
| | | SpinQuant* | 39.08 | 68.18 | 73.06 | 72.87 | 70.46 | 40.60 | 77.42 | 42.68 | 67.56 | 61.32 | 6.12 |
| | | OSTQuant* | 42.92 | 70.33 | 72.11 | 73.77 | 70.66 | 42.42 | 77.91 | 44.93 | 67.88 | 62.55 | 6.07 |
| 13B | 16-16-16 | Full Precision | 47.87 | 74.49 | 77.86 | 79.10 | 76.03 | 44.40 | 80.30 | 46.72 | 73.24 | 66.67 | 5.09 |
| | 4-4-16 | RTN | 25.85 | 26.26 | 42.05 | 26.70 | 0.17 | 28.00 | 50.33 | 34.60 | 50.67 | 31.63 | 2e4 |
| | | SmoothQuant | 25.43 | 29.29 | 51.56 | 28.12 | 2.02 | 26.00 | 53.32 | 34.34 | 49.57 | 33.29 | 6e2 |
| | | GPTQ | 24.66 | 27.78 | 40.80 | 25.83 | 0.0 | 24.20 | 51.31 | 36.65 | 51.70 | 31.51 | 3e3 |
| | | QuaRot | 48.04 | 72.18 | 76.02 | 76.90 | 72.56 | 43.60 | 79.43 | 44.47 | 70.24 | 64.83 | 5.57 |
| | | **DFRot** | 46.25 | 72.35 | 74.5 | 77.45 | 72.99 | 43.20 | 77.69 | 45.45 | 70.32 | 64.47 | 5.58 |
| | | SpinQuant* | 45.73 | 72.56 | 75.38 | 76.86 | 73.28 | 43.60 | 78.89 | 44.63 | 70.40 | 64.59 | 5.36 |
| | | OSTQuant* | 48.04 | 74.07 | 77.13 | 77.22 | 74.58 | 45.00 | 78.62 | 46.16 | 71.35 | 65.80 | 5.40 |
| | 4-4-4 | RTN | 26.28 | 27.27 | 42.35 | 25.85 | 0.19 | 26.60 | 49.95 | 34.19 | 49.25 | 31.33 | 3e4 |
| | | SmoothQuant | 24.49 | 28.83 | 51.65 | 27.91 | 2.08 | 26.00 | 52.56 | 35.41 | 50.59 | 33.28 | 5e2 |
| | | GPTQ | 23.63 | 27.31 | 39.85 | 26.17 | 0.56 | 26.00 | 51.96 | 35,82 | 49.57 | 30.63 | 3e3 |
| | | Omniquant | 29.61 | 48.23 | 58.20 | 56.45 | 28.76 | 31.40 | 65.29 | 37.10 | 55.64 | 45.63 | 10.87 |
| | | QuaRot | 47.10 | 70.92 | 75.6 | 76.31 | 72.44 | 44.8 | 77.91 | 45.65 | 71.35 | 64.68 | 5.59 |
| | | **DFRot** | 46.16 | 70.88 | 74.22 | 76.88 | 73.18 | 42.40 | 78.13 | 46.06 | 70.40 | 64.26 | 5.62 |
| | | SpinQuant* | 45.99 | 70.71 | 76.51 | 77.16 | 73.63 | 45.60 | 79.00 | 45.65 | 70.32 | 64.95 | 5.39 |
| | | OSTQuant* | 45.90 | 75.25 | 76.94 | 77.21 | 74.23 | 43.40 | 79.43 | 45.91 | 70.56 | 65.43 | 5.40 |
| 30B | 16-16-16 | Full Precision | 52.99 | 80.39 | 82.75 | 82.62 | 77.59 | 48.00 | 82.26 | 47.75 | 75.69 | 70.00 | 4.10 |
| | 4-4-16 | RTN | 25.00 | 27.95 | 42.02 | 27.22 | 0.21 | 27.00 | 49.13 | 34.65 | 50.91 | 31.57 | 2e3 |
| | | SmoothQuant | 23.63 | 30.68 | 54.86 | 31.91 | 3.80 | 28.20 | 54.13 | 34.49 | 50.04 | 34.64 | 1e3 |
| | | GPTQ | 27.30 | 27.19 | 38.69 | 26.75 | 0.17 | 25.80 | 49.02 | 35.21 | 47.75 | 30.88 | 2e3 |
| | | QuaRot | 52.05 | 75.84 | 81.28 | 80.78 | 75.33 | 46.2 | 80.36 | 45.91 | 72.38 | 67.79 | 4.74 |
| | | **DFRot** | 50.94 | 76.64 | 80.43 | 80.82 | 75.35 | 47.0 | 80.03 | 47.34 | 74.03 | 68.06 | 4.78 |
| | | SpinQuant* | 50.06 | 77.06 | 81.38 | 80.62 | 76.79 | 46.00 | 80.14 | 46.37 | 74.27 | 68.08 | 4.53 |
| | | OSTQuant* | 51.37 | 78.11 | 82.48 | 79.51 | 75.99 | 45.40 | 81.18 | 47.80 | 74.82 | 68.52 | 4.43 |
| | 4-4-4 | RTN | 25.00 | 28.87 | 44.07 | 27.29 | 0.39 | 25.60 | 49.67 | 34.54 | 49.33 | 31.64 | 2e3 |
| | | SmoothQuant | 22.61 | 32.87 | 55.05 | 31.28 | 3.40 | 28.00 | 53.75 | 34.65 | 50.28 | 34.65 | 1e3 |
| | | GPTQ | 27.22 | 27.82 | 39.36 | 27.13 | 0.33 | 24.80 | 50.71 | 34.34 | 47.91 | 31.07 | 2e3 |
| | | Omniquant | 29.10 | 53.79 | 54.95 | 52.44 | 26.45 | 30.60 | 65.56 | 38.54 | 53.91 | 45.04 | 10.33 |
| | | QuaRot | 51.96 | 76.22 | 80.21 | 80.96 | 74.52 | 46.40 | 79.87 | 46.67 | 74.51 | 67.92 | 4.77 |
| | | **DFRot** | 51.11 | 76.43 | 80.98 | 80.91 | 74.87 | 46.60 | 79.60 | 46.21 | 74.66 | 67.93 | 4.81 |
| | | SpinQuant* | 51.62 | 76.98 | 81.07 | 80.57 | 76.63 | 46.00 | 79.92 | 46.26 | 74.19 | 68.14 | 4.55 |
| | | OSTQuant* | 49.74 | 76.52 | 81.16 | 81.13 | 77.57 | 46.40 | 80.90 | 46.11 | 74.27 | 68.20 | 4.42 |

Table 6: Complete omparison of the perplexity score on WikiText2 and averaged accuracy on Zero-shot Common Sense Reasoning tasks on **LLaMA**.

| Model | #Bits W-A-KV | Method | ARC-c (↑) | ARC-e (↑) | BoolQ (↑) | HellaS. (↑) | LambA. (↑) | OBQA (↑) | PIQA (↑) | SIQA (↑) | WinoG. (↑) | Avg. (↑) | Wiki2 (↓) |
|---|---|---|---|---|---|---|---|---|---|---|---|---|---|
| 2-70B | 16-16-16 | Full Precision | 57.42 | 81.02 | 83.79 | 83.81 | 79.60 | 48.80 | 82.70 | 49.18 | 77.98 | 71.59 | 3.32 |
| | 4-4-16 | RTN | 29.35 | 26.05 | 37.74 | 25.97 | 0.02 | 24.80 | 51.31 | 34.14 | 48.70 | 30.90 | 7e4 |
| | | SmoothQuant | 25.00 | 35.98 | 55.23 | 32.52 | 7.49 | 25.00 | 54.62 | 35.21 | 51.70 | 35.86 | 3e2 |
| | | GPTQ | 27.82 | 25.80 | 37.95 | 25.82 | 0.00 | 27.00 | 49.67 | 33.98 | 49.72 | 30.86 | nan |
| | | QuaRot | 55.55 | 80.18 | 82.57 | 81.65 | 77.22 | 46.8 | 81.77 | 48.06 | 75.85 | 69.96 | 3.89 |
| | | **DFRot** | 55.46 | 78.83 | 82.32 | 79.19 | 76.87 | 46.40 | 81.23 | 46.21 | 75.93 | 69.16 | 3.99 |
| | | SpinQuant* | 55.38 | 78.96 | 83.36 | 82.54 | 79.00 | 47.80 | 82.10 | 48.67 | 77.43 | 70.58 | 3.68 |
| | | OSTQuant* | 56.61 | 80.51 | 83.03 | 82.68 | 79.11 | 47.86 | 83.00 | 48.76 | 76.70 | 70.92 | 3.57 |
| | 4-4-4 | RTN | 30.38 | 27.74 | 38.23 | 26.12 | 0.02 | 24.60 | 51.74 | 34.29 | 52.49 | 31.73 | 7e4 |
| | | SmoothQuant | 24.15 | 33.88 | 55.32 | 31.75 | 7.14 | 26.40 | 54.95 | 34.14 | 52.17 | 35.54 | 3e2 |
| | | GPTQ | 28.75 | 26.39 | 37.86 | 25.96 | 0.00 | 26.40 | 50.00 | 34.44 | 50.04 | 31.09 | nan |
| | | QuaRot | 55.55 | 79.55 | 82.02 | 81.39 | 78.03 | 47.2 | 80.79 | 47.49 | 77.58 | 69.96 | 3.92 |
| | | **DFRot** | 54.27 | 78.32 | 82.02 | 79.11 | 76.81 | 46.80 | 81.23 | 46.88 | 73.56 | 68.78 | 4.02 |
| | | SpinQuant* | 56.31 | 80.64 | 83.55 | 82.36 | 79.41 | 47.20 | 82.21 | 47.29 | 76.16 | 70.57 | 3.61 |
| | | OSTQuant* | 56.58 | 80.17 | 83.64 | 82.49 | 78.72 | 48.00 | 82.76 | 48.67 | 76.49 | 70.84 | 3.59 |
| 3-70B | 16-16-16 | Full Precision | 64.42 | 85.98 | 85.14 | 84.95 | 79.47 | 48.46 | 84.39 | 50.82 | 80.66 | 73.81 | 2.86 |
| | 4-4-16 | RTN | 27.47 | 25.88 | 37.83 | 26.26 | 0.00 | 27.20 | 51.63 | 35.26 | 49.33 | 31.21 | 9e3 |
| | | SmoothQuant | 25.60 | 34.47 | 50.46 | 32.48 | 1.98 | 30.00 | 54.24 | 33.83 | 48.93 | 34.67 | 2e2 |
| | | GPTQ | 25.77 | 26.09 | 43.64 | 26.42 | 0.00 | 27.40 | 52.01 | 32.55 | 49.33 | 31.47 | 4e4 |
| | | QuaRot | 52.99 | 76.52 | 81.59 | 79.35 | 75.1 | 47.8 | 79.87 | 46.37 | 74.66 | 68.25 | 5.92 |
| | | **DFRot** | 58.28 | 81.36 | 81.50 | 82.21 | 75.24 | 45.40 | 82.21 | 46.47 | 75.69 | 69.82 | 4.97 |
| | | SpinQuant* | 53.84 | 77.69 | 80.24 | 78.19 | 73.06 | 45.00 | 78.67 | 43.24 | 73.01 | 66.99 | 6.10 |
| | | OSTQuant* | 61.84 | 84.56 | 84.14 | 82.47 | 77.08 | 46.07 | 83.38 | 50.23 | 80.13 | 72.21 | 3.97 |
| | 4-4-4 | RTN | 27.13 | 25.42 | 37.83 | 26.12 | 0.00 | 26.60 | 50.76 | 35.16 | 48.38 | 30.82 | 9e3 |
| | | SmoothQuant | 23.46 | 31.48 | 48.81 | 29.22 | 4.13 | 28.00 | 52.56 | 34.95 | 51.22 | 33.76 | 3e2 |
| | | GPTQ | 26.11 | 25.17 | 45.17 | 26.07 | 0.00 | 26.40 | 48.86 | 33.88 | 49.17 | 31.20 | 4e4 |
| | | QuaRot | 53.75 | 77.4 | 80.46 | 79.37 | 75.39 | 45.40 | 79.82 | 46.42 | 76.64 | 68.29 | 6.02 |
| | | **DFRot** | 58.02 | 81.10 | 81.13 | 81.59 | 74.79 | 47.40 | 81.83 | 46.57 | 74.19 | 69.62 | 5.03 |
| | | SpinQuant* | 51.88 | 76.39 | 80.98 | 76.50 | 71.43 | 43.46 | 79.27 | 44.17 | 72.69 | 66.31 | 6.24 |
| | | OSTQuant* | 61.29 | 82.39 | 83.43 | 83.25 | 75.90 | 48.93 | 81.73 | 51.24 | 77.01 | 71.69 | 4.01 |

Table 7: Complete comparison of the perplexity score on WikiText2 and averaged accuracy on Zero-shot Common Sense Reasoning tasks for **LLaMA2-70B and LLaMA3-70B**.

