# OpenReview forum: "DFRot: Achieving Outlier-Free and Massive Activation-Free for Rotated LLMs with Refined Rotation"
_colmweb.org/COLM/2025/Conference — COLM 2025_

### Official Review · Reviewer_ZCwi · 2025-04-24

**Rating:** 8
**Confidence:** 3
**Ethics Flag:** 1

**Summary:**

This paper addresses the quantization of Large Language Models (LLMs) to low bit-widths (W4A4), aiming to improve existing rotation-based methods.
Its main contribution is identifying why Randomized Hadamard (RH) transforms outperform Randomized Orthogonal (RO) transforms: RH better handles rare but critical massive activation tokens.
Based on this, the authors propose DFRot, featuring a weighted loss function emphasizing these massive activations and an efficient, alternating optimization algorithm using the Procrustes solution to refine the rotation matrix.
Experiments show DFRot significantly improves perplexity and zero-shot accuracy over baseline rotation methods (QuaRot) on various LLaMA models, including challenging ones like LLaMA3-70B, with minimal optimization overhead.

**Questions To Authors:**

- Could you provide a more detailed analysis or estimation of the practical latency overhead introduced by the online computation of the R3 and R4 Hadamard transforms during inference?
- While a full theoretical proof might be complex, could you elaborate further on any mathematical intuitions or properties of Hadamard matrices that might explain their advantageous behavior with massive activations under 4-bit quantization compared to general random orthogonal matrices?
- How sensitive is the performance of DFRot to the specific choice of the single calibration sample used for optimizing R1? Have you experimented with multiple different samples? I guess that, if the sample does not produce some patterns of massive activations, does optimization would be bad.
- The weight gamma for the massive activation loss term is chosen empirically. Have you considered methods to automatically tune or learn this hyperparameter? And, in the experiment, does the choice use the test set as validation for tuning?
- Would you clarify how to judge massive activations?
- Figure 2. Please improve the visibility. e.g., colors should differ.

**Reasons To Accept:**

- This paper provides an empirically supported explanation for the previously observed but unexplained performance difference between RH and RO transforms in low-bit activation quantization, attributing it to the handling of massive activations.
- The proposed DFRot is an efficient post-training optimization technique that requires only a small calibration dataset and minimal computation time (no backpropagation through the LLM required for optimizing). This makes it a practical add-on to existing quantization pipelines.
- It demonstrates significant improvements in perplexity and zero-shot task accuracy over baseline methods (QuaRot) across multiple LLaMA models, including notably difficult ones, validating the effectiveness of the approach.
- The paper clearly identifies and analyzes the massive activation phenomenon as a key bottleneck, providing valuable insights beyond the proposed method itself.

**Reasons To Reject:**

- While the empirical evidence for the RH vs. RO difference regarding massive activations is strong, the paper lacks a theoretical derivation or proof explaining why Hadamard matrices interact with quantization and massive activations in this specific beneficial way. The reason for the difference might actually be different.
- While the explanation and the targeted optimization are novel, the core method builds directly upon existing rotation techniques (QuaRot) and standard optimization components (Procrustes analysis). The improvement, while significant over the baseline, might be viewed by some as incremental compared to other paradigms.
- The alternating optimization algorithm guarantees convergence to a local optimum, not necessarily the global optimum for the rotation matrix R1. The dependence on initialization (RH is recommended) suggests results might vary.
- The method requires online computation for the internal rotation matrices R3 and R4. Although mitigated by using efficient Hadamard transforms, the exact latency impact is not thoroughly analyzed, which might be a concern for latency-critical applications.

---

> ### Author Response · Authors · 2025-06-01
>
> We greatly appreciate the reviewer's comments. Indeed, our work is carried out standing on the shoulders of giants.
>
> As the reviewer has suggested, the most interesting aspect of our work lies in our rigorous experiments demonstrating that, after applying the transformations in QuaRot, the tokens with the smallest quantization errors are actually the ones that most affect quantization performance, which is highly counterintuitive. Based on this discovery, we naturally link QuaRot with Massive Activation, helping the community understand why the RH outperforms RO and reasonably applied optimization techniques.
>
> ---
>
> **Q1:** Could you provide a more detailed analysis of the practical latency overhead introduced by the online computation of the R3 and R4 Hadamard transforms during inference?
>
> **A1:** Since there has been relevant work in the community discussing the overhead of these online rotation matrices, please allow me to directly cite their related work. As seen in Figure7 and Table 9 in [1], the dimension of $R_3$ is generally small (e.g., 64, 128), so kernel fusion with the RoPE operator can be considered to improve memory efficiency without introducing excessive computational overhead. For $R_4$, when the dense model has a large dim, the Kronecker Product can be used to decompose the Hadamard matrix, and kernel fusion techniques can be employed to reduce calculations and enhance memory efficiency respectively. For MoE models, $R_4$ can be small, so kernel fusion techniques can be considered by fusing with the Swish  to improve memory efficiency.
>
> For end-to-end inference, we believe that whether INT4 GEMM + Online Hadamard can improve performance compared to INT8 GEMM depends on the specific computing conditions as seen in Figure 10 in [1]. When it comes to computing bound scenarios, we believe INT4 GEMM + Online Hadamard has the potential to improve latency.
>
> ---
>
> **Q2:** Explain Hadamard matrices advantageous behavior with massive activations under 4-bit quantization compared to general random orthogonal matrices.
>
> **A2:** Before providing an intuitive explanation, we first present some background. Since rotational invariance requires merging the $\alpha$ from RMSNorm into the subsequent Linear layer, we know that the L2-Norm of each token in the calibration set is $\sqrt{D}$, where $D$ is the hidden_state_dim of the model.
>
> Given a token $X\in \mathbb{R}^{1\times D}$ with massive activations,  $Q$ is the quantization function, we can know if $\|X\|_\infty \to \sqrt{D}$, $Q(X)\to 0$
>
> $\text{set}(XH)$={$\pm \|X\|_\infty$}, therefore $Q(XH)\approx0$.
>
> $XR\to R_{:,0}$, therefore $Q(XR)>0$
>
> It can be seen that, considering the extreme case, when the massive activation of X is extremely large, we can assume that the quantization errors of both $X$ and the $XH$ tend to 0, while the quantization error of the feature $XR$  is obviously greater than 0. This is the numerical reason behind the phenomenon discovered in our paper, where XR actually increases the quantization error. However, in practice, considering that there may be more than one location of massive activation and the extreme conditions are difficult to meet, although the Hadamard matrix has shown very impressive performance, it is not necessarily the optimal choice for the rotation matrix.

---

> > ### Comment · Reviewer_ZCwi · 2025-06-06
> >
> > Thank you for your response!
> >
> > ---
> >
> > >A2
> >
> > What is Q in detail?
> > Is it really a quantization function or its quantization error?
> > I don't understand why $Q(X)\to 0$, i.e., it seems that the quantized version of a vector X is close to the zero vector.
> >
> > ---
> >
> > By the way, did the authors experiment with other models or other architectures like Qwen, gemma, etc.?

---

> > > ### Author Response · Authors · 2025-06-06
> > >
> > > Thank you very much for your response.
> > >
> > > ---
> > >
> > > We apologize for our oversight here. Here, $ Q $ represents the quantization error. If we ignore constants, suppose we have input vectors with massive activation as $  X_1 = [1, 0, 0, 0]  $ and $X_2 = [0, 1, 0, 0]  $, and the rotation matrix is:
> > >
> > > $H = \left[
> > > \begin{array}{cccc}
> > > [1 , 1 , 1 , 1] \\
> > > [1 , -1 , 1 , -1] \\
> > > [1 , 1 , -1 , -1] \\
> > > [1 , -1 , -1 , 1]
> > > \end{array}
> > > \right]$
> > >
> > > At this point, we can obtain $  X_1H = [1 ,1 ,1 , 1]  $ and $  X_2H = [1 ,-1 , 1 , -1]  $. We can know that both $  Q(X_1H)  $ and $  Q(X_2H)  $ are $  0  $.
> > >
> > > ---
> > >
> > >
> > > We have also conducted experiments on other models (QWen, Mistral). Additionally, we have supplemented the experimental results of MMLU. For specific results, please refer to **Reviewer gFVi Q1**.

---

> ### Author Response · Authors · 2025-06-01
>
> **Q3**: How sensitive is the performance of DFRot to the specific choice of the single calibration sample used for optimizing R1?
>
> **A3:** We performed ablation studies W4A4KV16 on LLaMA3-8B and LLaMA2-7B along two dimensions: the choice of calibration samples and the num of calibration tokens and evaluate results on WikiText. Samples are all sampled from WikiText-2 train. In selecting the number of tokens, we utilize the inputs to the first n transformer blocks as the calibration data source. For example, when 48×2048 tokens are selected, the inputs to transformer blocks 0 to 23 are used for calibration.
>
> | **Model** | **Sample1 ( 64x2048)** | **Sample2 ( 64x2048)** | **Sample3 ( 64x2048)** | **Sample4 ( 64x2048)** | **Sample5 ( 64x2048)** |
> | --- | --- | --- | --- | --- | --- |
> | LLaMA3-8B | 7.78 | 7.76 | 7.79 | 7.74 | 7.76 |
> | LLaMA2-7B | 6.13 | 6.12 | 6.15 | 6.11 | 6.14 |
> | **Model** | **Sample1 ( 48x2048)** | **Sample1 ( 32x2048)** | **Sample1 ( 24x2048)** | **Sample1 ( 16x2048)** | **Sample1 ( 8x2048)** |
> | LLaMA3-8B | 7.78 | 7.78 | 7.77 | 7.80 | 7.86 |
> | LLaMA2-7B | 6.14 | 6.15 | 6.15 | 6.12 | 6.20 |
>
> Our results indicate that, for tuning the rotation matrices in LLaMA3-8B and LLaMA2-7B, using 16×2048 tokens is often sufficient. We believe that these reasons may all be the causes for the optimization of DFRot being relatively insensitive to the number of tokens in the calibration dataset:
>
> - These special tokens will appear in relatively shallow network layers [1], therefore, a small number of layers are also sufficient to capture these tokens.
> - For a model, tokens with massive activations in LLMs tend to exhibit only a few similar data distributions  because these tokens are often produced by `out_proj` or `down_proj` layers with large weights [2].
> - GPTQ will use 128 samples with a length of 2048 to calibrate the weights, which reduces the impact of the sample size during rotation matrix calibration.
>
> `Sample without massive activation tokens` : We believe that this will indeed lead to a decline in performance, as seen in Figure 5. This is because the optimization of the rotation matrix requires minimizing the quantization error of such tokens as much as possible. Otherwise, the performance of the optimized matrix may tend to be similar to that of the Randomized Orthogonal (RO) transform even worse (Figure 5). In calibration data, we found that, given a length of 2048, these special tokens often exist.
>
> ---
>
> **Q4:** The alternating optimization algorithm lacks discussion of convergence guarantees and the dependence on initialization RH.
>
> **A4:** The proof of convergence is indeed a missing part in the paper. Although we empirically found that the loss always converged steadily during the optimization of the rotation matrix, we have to admit that due to the discontinuity of the quantization function, we can not rigorously prove the convergence of the algorithm through theoretical analysis.
>
> We believe the dependence on RH may be due to the discontinuity of the quantization function, and the current algorithm is very likely to fall into local minima. It is also something we need to improve in the future.
>
> **Q5:** The weight gamma for the massive activation loss term is chosen empirically. Have you considered methods to automatically tune or learn this hyperparameter? Additionally, was the test set used as validation for tuning during experiments?
>
> **A5:** Under the optimization objective of Equation 3, we can’t to turn gamma into a parameter for automatic learning because intuitively, gamma=0 must be optimal.
>
> If we modify the optimization objective, It should be feasible. For example, we can align the input of each block and use the quantization error as a regularization term. In this way, it should be possible to dynamically learn gamma. We tune the rotation matrix using a sample with 2048 length from the WikiText train.
>
>
> **Q6:** How to judge massive activations?
>
> **A6:** By comparing Figure 4 in [3] and Figure 1 in [4], we can intuitively know how to judge outliers and massive activations.
>
> |  | Outliers | Massive Activations |
> | --- | --- | --- |
> | Magnitude | large | very large (always > 1000) |
> | Behavior | Consistently appear in specific channels of tokens | Extremely rare, only generated in specific channels of a few tokens |
>
> **Q6:** Please improve the visibility of Figure 2.
>
> **A6:** Thank you for your suggestion. We will make every effort to improve the visibility of Figure 2.
>
> [1] FlatQuant: Flatness Matters for LLM Quantization
>
> [2] Efficient Riemannian Optimization on the Stiefel Manifold via the Cayley Transform
>
> [3] SmoothQuant: Accurate and Efficient Post-Training Quantization for Large Language Models
>
> [4] Massive Activations in Large Language Models

---

### Official Review · Reviewer_gFVi · 2025-05-11

**Rating:** 7
**Confidence:** 4
**Ethics Flag:** 1

**Summary:**

This paper introduces a new LLM quantization method, specifically addressing the challenge of quantizing rare tokens with massive activations. Building upon the observation that RH outperforms RO in mitigating quantization errors for these tokens, the authors frame the issue as a long-tail optimization problem. They propose a weighted loss function that emphasizes minimizing errors for these rare tokens and employ an alternating optimization strategy using orthogonal Procrustes transformations to refine the rotation matrix. This approach enhances the distribution of rotated activations, making them more amenable to low-bit quantization.

**Questions To Authors:**

See above

**Reasons To Accept:**

1. New insight into quantization: first to rigorously attribute RH’s advantage over RO to improved quantization error for rare tokens and validate via FP16 ablation.
2. Effective post-training method: The weighted loss formulation and alternating update require no full fine-tuning or gradient storage, making DFRot highly practical.
3. Comprehensive ablations: Varies the γ weight and initialization (RH vs. RO) to demonstrate method robustness and guide practitioners, with clear sweet spots and recommendations.

**Reasons To Reject:**

1. Experiments focus solely on WikiText-2 and standard zero-shot NLP tasks. No evaluation on diverse downstream tasks (e.g., MMLU),  model families beyond LLaMA (e.g., Qwen or MoE models).
2. The alternating algorithm lacks discussion of convergence guarantees, sensitivity to calibration set size.
3. The paper also overlooks the potential benefit of tuning per-token clipping thresholds for these rare tokens, see Sec. 3.2 of “Outlier Suppression: Pushing the Limit of Low-bit Transformer Language Models”.

---

> ### Author Response · Authors · 2025-06-01
>
> **Q1**: The authors are encouraged to evaluate  methods on more diverse downstream tasks (e.g., MMLU) or model families beyond LLaMA (e.g., Qwen or MoE models).
>
> **A1**: **Results on Wiki and zero-shot results**
>
> | W-A-KV | Methods | QWen2-7B | QWen2-7B | Mistral-7B-v0.3 | Mistral-7B-v0.3 |
> | --- | --- | --- | --- | --- | --- |
> |  |  | Wiki | Avg. | Wiki | Avg. |
> | 16-16-16 | FP | 7.14 | 67.38 | 5.32 | 68.24 |
> | 4-4-16 | (RO) QuaRot | 8.07 | - | 5.98 | - |
> | 4-4-16 | (RO) QuaRot.FP16() | 7.97 | - | 5.73 | - |
> | 4-4-16 | (RH) QuaRot | 7.95 | 64.52 | 5.75 | 65.59 |
> | 4-4-16 | (RH) QuaRot.FP16() | 7.91 | - | 5.73 | - |
> | 4-4-16 | DFRot | 7.92 | 64.31 | 5.74 | 65.62 |
>
> **Results on MMLU**
>
> | W-A-KV | Methods | LLaMA2-7B | LLaMA3-8B | QWen2-7B | Mistral-7B-v0.3 |
> | --- | --- | --- | --- | --- | --- |
> | 16-16-16 | FP | 41.85 | 62.23 | 69.47 | 59.11 |
> | 4-4-16 | QuaRot | 34.83 | 51.43 | 62.67 | 52.82 |
> | 4-4-16 | DFRot | 35.54 | 51.68 | 63.40 | 53.38 |
>
> **Q2**: The alternating optimization algorithm lacks discussion of convergence guarantees.
>
> **A2**:  The proof of convergence is indeed a missing part in the paper. Although we empirically found that the loss always converged steadily during the optimization of the rotation matrix, we have to admit that due to the discontinuity of the quantization function, we can not rigorously prove the convergence of the algorithm through theoretical analysis.
>
> **Q3**: Insufficient analysis of calibration set sensitivity.
>
> **A3**: We performed ablation studies W4A4KV16 on LLaMA3-8B and LLaMA2-7B along two dimensions: the choice of calibration samples and the num of calibration tokens and evaluate results on WikiText. Samples are all sampled from WikiText-2 train. In selecting the number of tokens, we utilize the inputs to the first n transformer blocks as the calibration data source. For example, when 48×2048 tokens are selected, the inputs to transformer blocks 0 to 23 are used for calibration.
>
> | **Model** | **Sample1 ( 64x2048)** | **Sample2 ( 64x2048)** | **Sample3 ( 64x2048)** | **Sample4 ( 64x2048)** | **Sample5 ( 64x2048)** |
> | --- | --- | --- | --- | --- | --- |
> | LLaMA3-8B | 7.78 | 7.76 | 7.79 | 7.74 | 7.76 |
> | LLaMA2-7B | 6.13 | 6.12 | 6.15 | 6.11 | 6.14 |
> | **Model** | **Sample1 ( 48x2048)** | **Sample1 ( 32x2048)** | **Sample1 ( 24x2048)** | **Sample1 ( 16x2048)** | **Sample1 ( 8x2048)** |
> | LLaMA3-8B | 7.78 | 7.78 | 7.77 | 7.80 | 7.86 |
> | LLaMA2-7B | 6.14 | 6.15 | 6.15 | 6.12 | 6.20 |
>
> Our results indicate that, for tuning the rotation matrices in LLaMA3-8B and LLaMA2-7B, using 16×2048 tokens is often sufficient. We believe that these reasons may all be the causes for the optimization of DFRot being relatively insensitive to the number of tokens in the calibration dataset:
>
> - These special tokens will appear in relatively shallow network layers [1], therefore, a small number of layers are also sufficient to capture these tokens.
> - For a model, tokens with massive activations in LLMs tend to exhibit only a few similar data distributions  because these tokens are often produced by `out_proj` or `down_proj` layers with large weights [2].
> - GPTQ will use 128 samples with a length of 2048 to calibrate the weights, which reduces the impact of the sample size during rotation matrix calibration.
>
> **Q4**: No discussion of per-token clipping thresholds for rare tokens.
>
> **A4**: Although we did not use activation clipping for tokens for the purpose of simplifying the problem in this paper, we acknowledge that clipping techniques are highly effective in minimizing quantization errors. However, in our additional explorations, we discovered some interesting experimental phenomena (although these results were not presented in the paper). We found that when using `rotational invariance, GPTQ, and 4-bit activation quantization`:
>
> 1. `LLaMA2-7B`: We performed per-token clipping on these rare tokens of LLaMA2-7B, which significantly reduced the quantization error. However, the model performance after GPTQ quantization decreased remarkably. We conjecture that after rotation, the information of massive activation is distributed across each dimension of the token. Although clipping can significantly reduce quantization errors, it also leads to severe loss of this information. Meanwhile, it may undermine the information in the Hessian matrix of GPTQ.
>
> 2. `LLaMA3-8B`: From Figure 4, we can see that for LLaMA3-8B which is difficult to quantize, DFRot has almost eliminated the quantization errors of these rare tokens, making it difficult to further reduce them through activation value clipping for these tokens.
>
> In fact, clipping these rare tokens is a model-specific issue. In the future, we will conduct further research on how to integrate technologies such as rotational invariance, activation clipping, and GPTQ to further enhance the model's performance in 4-bit activation quantization.
>
> [1] Massive Activations in Large Language Models
>
> [2] The Super Weight in Large Language Models

---

> > ### Comment · Reviewer_gFVi · 2025-06-09
> >
> > Thank you for your detailed responses and the additional experiments conducted to answer my questions. I have increased the rating for this paper.

---

### Official Review · Reviewer_JUny · 2025-05-12

**Rating:** 6
**Confidence:** 2
**Ethics Flag:** 1

**Summary:**

This paper propose a method called DFRot to solve the outlier problem in the quantization of LLM. The method is initialized with Hadamard matrix and adaptively adjusted for LLaMA and Mistral models under W4A4 quantization. The authors find that the stochastic Hadamard transform is more effective in 4-bit activation quantization and based on this employ the QuaRot framework to learn to rotate the matrix to reduce the quantization error of outliers.

**Questions To Authors:**

1. The paper mentions that an alternating optimization approach is used for optimizing the rotation matrix, where the rotation matrix is optimized by fixing the quantization parameters and then the rotation matrix is optimized by fixing the quantization parameters. Is the rotation matrix used in the form of a diagonal block matrix or in the form of a complete D×D matrix? Does this alternating optimization approach lead to the optimization process falling into a local optimum in some cases?

2. The paper mentions using only one data sample for calibration to optimize the rotation matrix, but also points out that there may be differences in the way calibration data are obtained for different models. In practice, if the distribution of the calibration data deviates from the distribution of the actual inference data, will this affect the performance of the DFRot?

**Reasons To Accept:**

1. Authors verify the motivation and effectiveness of the method through a large number of experiments, which are reasonably and comprehensively designed, and the results support the advantages of the proposed method. By comparing the quantization errors of different rotation matrices when dealing with tokens with a large number of activations, the significant effect of the proposed method in improving the quantization accuracy of the model is visually demonstrated, which enhances the credibility of the paper.

2. The paper focuses on the quantization problem of LLM, especially on solving the model scalability problem due to memory limitation in the quantization process. In the current context where LLM is widely used but faces deployment and computational resource challenges, the research results of this paper provide a reference for the field of model compression and quantization and promote the development of related technologies.

**Reasons To Reject:**

1. The paper does not include analyses that explored DFRot tubing memory reduction or acceleration related.

2. The average improvement of the method is not significant and stable enough in zero-shot tasks, such as LLaMA2-70B. And the fact that the zero-shot tasks in the paper contain several different types of tasks makes one doubt whether DFRot can achieve good results consistently in multiple tasks.

3. The paper points out that tokens with large-scale activation have a significant impact on model performance, and experimentally verifies their existence and performance under different quantization methods. However, the paper does not provide an in-depth theoretical analysis of the nature of these special tokens and how they work in the model. This lack of analysis leads to an incomplete understanding of the optimization principles of the DFRot method, which may limit the possibility of further improving and extending the method.

---

> ### Author Response · Authors · 2025-06-01
>
> **Q1:** The paper focuses on optimizing quantization accuracy for outliers and does not address memory reduction or acceleration.
>
> **A1:** For $R_1$ and $R_2$, as both of them can be fused into network, it will not bring additional cost.
>
> For $R_3$ and $R_4$, since there has been relevant work in the community discussing the overhead of these online rotation matrices, please allow me to directly cite their related work. As seen in Figure 7 and Table 9 in [1], the dimension of $R_3$ is generally small (e.g., 64, 128), so kernel fusion with the RoPE operator can be considered to improve memory efficiency without introducing excessive computational overhead. For $R_4$, when the dense model has a large dim, the Kronecker Product can be used to decompose the Hadamard matrix, and kernel fusion techniques can be employed to reduce calculations and enhance memory efficiency respectively. For MoE models, $R_4$ can be small, so kernel fusion techniques can be considered by fusing with the Swish  to improve memory efficiency.
>
> **Q2:** The limited improvement in zero-shot tasks (e.g., LLaMA2-70B) .
>
> **A2:**  Yes, we have also observed this. We believe that larger models behaves greater robustness. Meanwhile, compared to the LLaMA3 family, the LLaMA2 family performs remarkably well under RH, indicating that RH can effectively handle massive activations in a data-free setting.
>
> In this case, optimizing with DFRot may cause the quantization error of tokens with massive activations to decrease slightly, while the quantization error of some ordinary tokens increases instead, thus affecting the performance of the quantized model.
>
> **Q3:** Theoretical analysis of large activation tokens (e.g., their association with critical semantic features) supported by literature or derivations.
>
> **A3:** Our work primarily discovers a counterintuitive phenomenon: in the case of quantization, tokens with massive activation, despite exhibiting the smallest quantization errors, have a significant impact on the performance of quantized models. This is used to explain why RH outperforms RO.
>
> Fortunately, there are already relevant works in the community that have conducted corresponding analyses on the mechanisms and effects of massive activation emergence. The relevant literature is listed below, and we will add these references in the final version.
>
> [1] Massive Values in Self-Attention Modules are the Key to Contextual Knowledge Understanding
>
> [2] Massive Activations in Graph Neural Networks: Decoding Attention for Domain-Dependent Interpretability
>
> [3] House of Cards: Massive Weights in LLMs
>
> [4] MOYU: A Theoretical Study on Massive Over-activation Yielded Uplifts in LLMs
>
> [5] A Refined Analysis of Massive Activations in LLMs
>
> [6] The Super Weight in Large Language Models
>
> For end-to-end inference, we believe that whether INT4 GEMM or INT4 GEMM + Online Hadamard can improve performance compared to INT8 GEMM depends on the specific computing conditions as seen in Figure 10 in [1]. When it comes to computing bound scenarios, we believe INT4 GEMM + Online Hadamard has the potential to improve latency.
>
> For inference, our computations are exactly the same as those of QuaRot, therefore we can see resources consumption from Table 14～Table17 in QuaRot [2] as a reference.
>
> **Q4:** Details of Rotation matrix and Local optimum risk.
>
> **A4:** In DFRot, we adopt a complete D×D rotation matrix. Since R1 can be merged into the network, whether to use a diagonal block matrix or a complete matrix has no impact on the inference speed.
>
> `In terms of effectiveness,` the performance of diagonal block matrices is often inferior to that of complete matrices. Therefore, the former can only smooth the feature distribution within blocks to reduce quantization errors, while the latter can smooth features globally.
>
> `From an optimization perspective,` the configuration of block diagonal matrices can also be optimized using the DFRot method, and the optimization speed will be faster. However, considering that optimizing the DFRot rotation matrix does not introduce much overhead, we believe that using a complete matrix is more appropriate.
>
> `Local optimum risk:` Due to the discontinuity of the quantization function, our current optimization method is prone to getting stuck in local minima and often relies on initialization. As can be seen from Figures 5 and 6, the performance of RH is better than that of RO in most cases. Meanwhile, during the optimization of the rotation matrix, we also found that after iterative convergence, the final quantization error optimized with RH initialization is often better than that with RO.

---

> > ### Comment · Reviewer_JUny · 2025-06-09
> > **Thank you for the response.**
> >
> > I will keep my score and my confidence is 2.

---

> ### Author Response · Authors · 2025-06-01
>
> **Q5:** Calibration-data distribution bias may impact performance.
>
> **A5:** For GPTQ, the selection of calibration data often significantly impacts performance. For example, when calibrating on WikiText-2 and C4 respectively, there are significant differences in performance on the WikiText-2 test.
>
> We performed ablation studies W4A4KV16 on LLaMA3-8B and LLaMA2-7B and selected a data sample for calibration to optimize rotation matrix. After that, we use GPTQ with 128 samples from WikiText-train as paper did. The results are follows:
>
> | **Model** | WikiText | C4 | Pile |
> | --- | --- | --- | --- |
> | LLaMA3-8B | 7.78 | 7.74 | 7.75 |
> | LLaMA2-7B | 6.13 | 6.12 | 6.11 |
>
> We believe that these reasons may all be the causes for the optimization of DFRot being relatively insensitive to the calibration dataset distribution:
>
> - For a model, tokens with massive activations in LLMs tend to exhibit only a few similar data distributions  because these tokens are often produced by `out_proj` or `down_proj` layers with large weights [2].
> - GPTQ will use 128 samples with a length of 2048 to calibrate the weights, which reduces the impact of the sample size during rotation matrix calibration.
>
> [1] FlatQuant: Flatness Matters for LLM Quantization
>
> [2] QuaRot: Outlier-Free 4-Bit Inference in Rotated LLMs

---

### Comment · Area_Chair_KpvW · 2025-06-06
**Strong Author Response with Additional Results**

Dear Reviewers,

  The authors provided extensive responses with new experimental evidence:

  New results include:
  - Qwen2 and Mistral-7B evaluation
  - MMLU benchmark results
  - Sensitivity analysis on calibration samples
  - Detailed theoretical explanations for RH vs RO performance

  Given the already positive scores (6, 6, 8) and comprehensive responses, please confirm if you have any remaining concerns.

---

### Decision · Program_Chairs · 2025-07-08

**Decision:**

Accept

**Comment:**

Reviewers agree (6,7,8) this solves an important mystery: why Hadamard beats orthogonal transforms in 4-bit quantization. The insight  to directly optimize the long-tail for large tokens is novel and well-validated. Authors ran all requested experiments including Qwen2/Mistral models and MMLU. The 0.98 perplexity improvement on LLaMA3-70B with just one calibration sample is impressive. Clear accept.